# iMOE: prediction of second-life battery degradation trajectory using interpretable mixture of experts

Xinghao Huang[1,8], Shengyu Tao [1,2,3,8] ✉, Chen Liang [1], Yining Tang [4], Jiawei Chen[5], Junzhe Shi [2], Yuqi Li [6], Bizhong Xia[1] ✉, Guangmin Zhou [1] ✉ & Xuan Zhang [1,7] ✉

Retired electric vehicle batteries offer immense potential to support energy infrastructure stability in underdeveloped regions through second-life use, but uncertainties in battery degradation behaviors pose major safety concerns. This work proposes an interpretable mixture of experts (iMOE) network that predicts battery degradation trajectories using partial, field-accessible signals in a single cycling operation. iMOE leverages an adaptive multi-degradation prediction module to classify battery degradation modes using expert weight synthesis learned from battery capacity-voltage and relaxation data. The module produces latent degradation trend embeddings, which are input to a use-dependent recurrent network for long-term degradation trajectory prediction. Validated on three typical use patterns (i.e. consistent operating histories, deeply aged batteries with unknown prior use, and uncertain second-life conditions, including 295 batteries, 93 use conditions, and 84,213 cycles), iMOE achieves an average mean absolute percentage errors (MAPE) of 0.95% with a 0.43 ms inference time for life-long battery degradation trajectory prediction. Compared to state-of-the-art Informer and PatchTST, it reduces computational time and MAPE by 50% and 77%, respectively. Compatible with data sampling in random state of charge regions, iMOE supports a 150-cycle time-horizon degradation trajectory prediction with 1.50% and 6.26% MAPE on average and at maximum, respectively. Notably, iMOE can operate effectively even with pruned 5MB training data while retaining 0.95% MAPE. Broadly, this network offers a deployable, history-free solution for battery degradation trajectory prediction at the time of second-life deployment, redefining how second-life energy storage systems are sensed, evaluated, controlled, and integrated for sustainable energy infrastructures at scale.

Lithium-ion batteries have been the core energy storage medium for electric vehicles (EVs) and renewable energy systems[1]. However, it is projected that by 2030, 120 GWh of in-service batteries will reach end of life (EOL) and be retired from primary use, such as EVs, bringing considerable economic and environmental concerns[2,3]. The reusing and recycling of these retired EV batteries has been identified emerging solution to improve the affordability and sustainability in battery lifecycle, particularly for the energy infrastructures in underdeveloped regions[4–7].

A full list of affiliations appears at the end of the paper. ✉e-mail: shengyu.tao@chalmers.se; xiabz@sz.tsinghua.edu.cn; guangminzhou@sz.tsinghua.edu.cn; xuanzhang@sz.tsinghua.edu.cn

Degradation trajectory is a widely adopted indicator for battery degradation characterization, which can be critical to the safety performance during extended reusing or repurposing process, i.e., second-life of the retired batteries[8–11]. For example, retired batteries undergo degradation trajectory evaluation before second-life use based on the cell, module or pack-level application requirements[12,13]. Conventional non-data-driven methods relying on destructive disassembly or full-cycle capacity tests[14,15], inflicting extra damage or labor investments. Notably, the economic feasibility of these state measurement methods of retired batteries are impractical due to prohibitive time, momentary, and environmental costs at scale[1,16–18]. Data-driven methods have demonstrated promises in predicting degradation trajectories using non-destructive electrical signal data, but they typically require 5–40% of full lifecycle historical data[19–22]. Moreover, uncertain second-life use conditions can significantly differ from those in first-life, rendering conventional battery degradation trajectory prediction models ineffective, given that the data used for model training were built on constant-condition tests or typical dynamical cycling tests while the degradation trajectory is highly path-dependent[23–25]. Lu et al. leveraged at least one full cycle of capacity-voltage curves to predict degradation trajectories under uncertain future operating scenarios by learning the relationship between use condition and battery capacity[23]. Yet, the method encounters limitations due to the stochasticity in state of charge (SOC) of retired batteries and demanding requirement for full discharge-charge data. Moreover, the unavailability of historical cycling data of retired batteries complicates the tracing of initial conditions of degradations, such as impedance rise, loss of lithium inventory, and loss of active material[26–29], challenging degradation trajectory prediction effectiveness. At the policy level, global initiatives such as the Battery Data Genome Initiative[30] and the European Commission's July 4, 2025, Delegated Regulation on battery recycling[31] aim to enhance data standardization, traceability, and material recovery across the battery lifecycle. However, policy advances still face practical challenges due to frequent ownership change and uncertain second-life use conditions of retired batteries, emphasizing the need for degradation prediction frameworks that rely solely on field-accessible data and adapt to diverse operational environments.

Additionally, due to highly non-linear degradation of second-life batteries, the incorporation of physics knowledge into models has gained increased attention. Recent advances in sensory-based measurements include X-ray imaging[32], electrochemical impedance, optical fiber sensing[33], acoustic sensing[34], and partial charging[35], which serve as the data input of the data-driven methods. Nevertheless, most sensing techniques remain at the laboratory stage and are invasive[33]. Meanwhile, purely data-driven methods struggle to capture internal physical information of retired batteries[36]. Given the understanding that internal physical state is particularly critical for second-life applications with safety-sensitive considerations, the challenge of the "black-box" nature and lack of interpretability in data-driven methods for battery state prediction should be addressed. Even if progresses have been made in physics-informed neural networks[37,38], the promise lies in additional and explicit integration of physical laws into feature engineering process[39,40], neural network loss functions[36], and transferability metrics[41]. However, the complex physical constraints involve numerous parameters and prior physical understanding from the modeling of retired batteries, which are still hardly available post-retirement[12]. Tao et al. employed physics-informed machine learning to achieve full degradation trajectory prediction using early-cycle data, reducing data requirements, while still relying on historical data[22]. In recent years, the mixture-of-experts (MOE) architecture, a core component in large language models, has demonstrated notable performance in learning heterogeneous data representations. By decomposing complex tasks into sub-tasks handled by "expert" modules with specialized task knowledge, MOE captures multi-level and nonlinear feature representations. The ideal of decomposing tasks into multiple subtasks naturally aligns with coupled degradation mechanisms of batteries. However, in absence of physical interpretability, the learning outcomes of model are confined to purely statistical correlations and fail to reflect true physical mechanisms underlying degradation processes, particularly with diverse cathode chemistries[42], historical usages[43], and uncertain second-life operating conditions[23]. Incorporating physical knowledge into the MOE architecture anchors data-driven learning outcomes to typical electrochemical processes, thereby enhancing model interpretability and enabling mechanism-aware degradation trajectory prediction.

To fulfil this gap, this work proposes an interpretable mixture of experts (iMOE) network for predicting degradation trajectories of retired batteries under uncertain second-life use complexities without requiring historical data. As shown in Fig. 1a, the model uses electrical signal data from randomly sampled SOC regions at collection field of retired batteries, interactively incorporating assumed future use conditions to predict degradation trajectories toward extended time horizons in future second-life use. Figure 1b illustrates the designed adaptive multi-mode degradation prediction (AMDP) module, which integrates multiple expert systems' weights studied from capacity-voltage and relaxation data to achieve classification of degradation modes of retired batteries. AMDP outputs latent degradation trend embedding vectors that are forwarded into a feature-operational recurrent neural network (FORNN), enabling interpretable and extendable prediction of degradation trajectories. In Fig. 1c, model's applications on battery reusing and recycling are illustrated, highlighting a safety-aware allocation of second-life batteries based on their predicted degradation trajectories toward a sustainable battery circular economy. Without historical data, the proposed approach achieves an average mean absolute percentage errors (MAPE) of 0.88% in predicting degradation trajectories under uncertain and diversified second-life use scenarios, spanning 295 batteries and 93 use conditions and 84,213 cycles. Notably, the required input data can be easily extracted at random SOC region, without charging or discharging retired batteries to anticipated SOC levels. The proposed method can extend degradation trajectory prediction over a 150 cycle of horizon in future second-life use with an MAPE of 1.50%. This work underscores transformative potential of incorporating physics prior principles into predictive health management of retired batteries, enabling broader proactive fault detection and reliability assessment across complex, safety-critical systems, such as large-scale energy infrastructures, with careful consideration of durability, safety, and sustainable operation under data-scarce and data-heterogeneous conditions.

## Results
### Methodology overview
The objective of this study is to compute battery degradation trajectories under uncertain future use conditions using one cycle data collected at random SOC regions. We use partial charging curves, relaxation voltage, and future conditions (charge/discharge current, temperature) as required model inputs (specific extraction methods are detailed in "Methods"). Input signal selection prioritizes two critical aspects: first, the controllability of charging processes in second-life battery ensures easier signal acquisition; second, relaxation voltage measurements remain unaffected by charge/discharge operations, thus enhancing model applicability. Six physics-informed features extracted from both relaxation voltage and partial charging curves (detailed in Supplementary Figs. 1–6 and Supplementary Notes 1–2) characterize polarization and degradation states. These features drive the model to classify batteries with different aging modes into distinguishable latent subspaces. Notably, physics-informed feature extraction extends beyond the charging and relaxation voltage curves discussed here but serves as the inputs of the AMDP module.

A two-phase collaborative mechanism achieves deep integration of physical insights and data-driven approaches: (1)

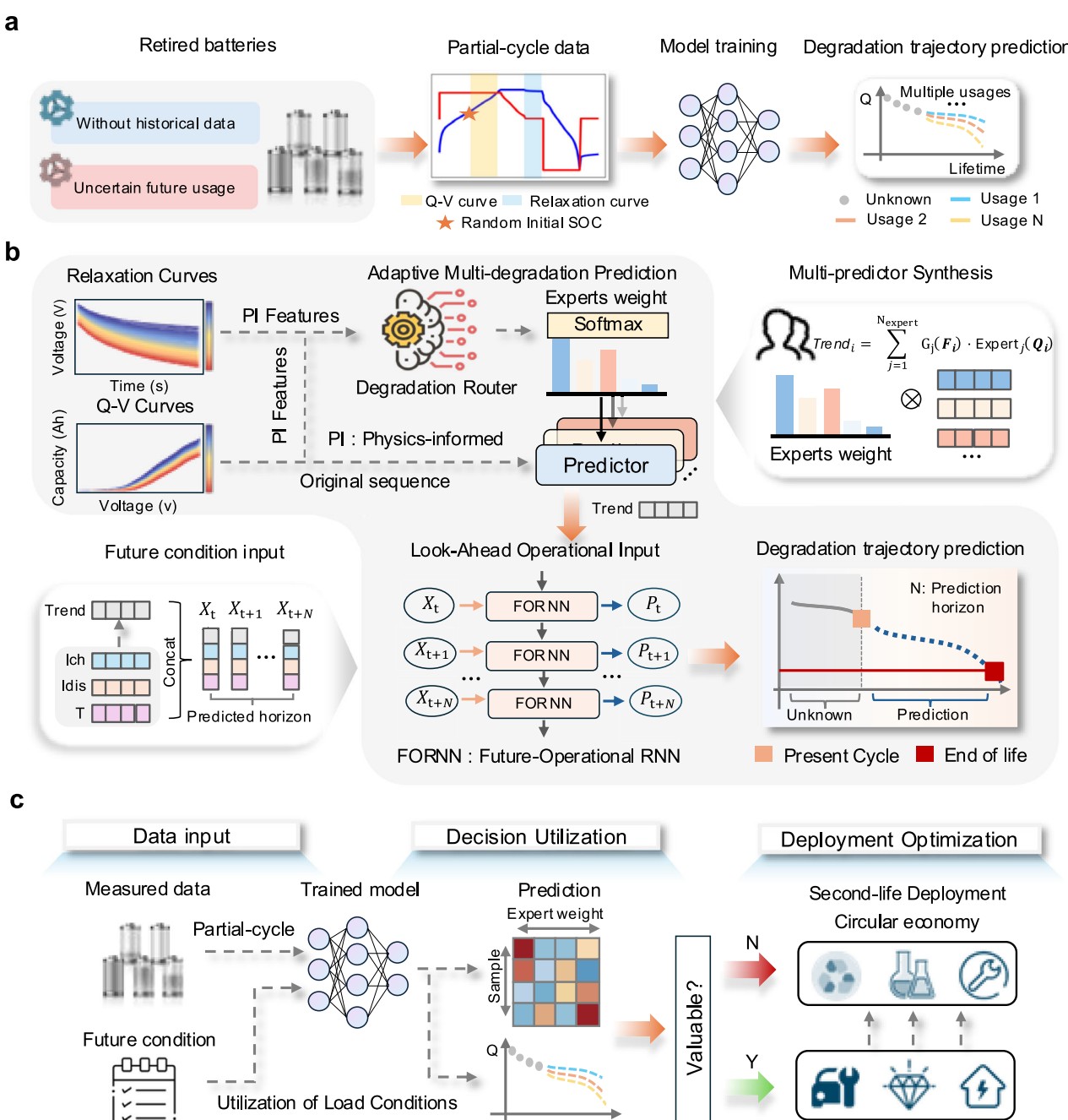

**Fig. 1 | Model motivation, model architecture, and model deployment. a** Using field-accessible partial cycle data with random initial SOC to predict retired battery degradation trajectories under conditions of unknown historical data and uncertain secondary use conditions. **b** The iMOE consists of two modules: AMDP and FORNN. The AMDP module uses physical features extracted from relaxation voltage and random initial SOC charging curves to guide the degradation router in adaptively integrating predictions from multiple degradation modes to output degradation trends. The FORNN module combines the degradation trends predicted by the AMDP module with future-operational to predict future degradation trajectories. **c** The trained model can utilize the output expert network to perform value assessment and preliminary utilization decisions for retired batteries and conduct economically and temporally reasonable secondary utilization deployment and recycling.

AMDP, and (2) FORNN. The AMDP module employs "degradation-router" routing to assign expert network weights based on the physical features, adaptively modeling degradation modes under dominant mechanisms for degradation trend representation[44], which can be formulated as:

$$Trend_i = \sum_{j=1}^{N_{expert}} G_j(\boldsymbol{F}_i) \cdot Expert_j(\boldsymbol{Q}_i) \qquad (1)$$

where, $Trend_i$ denotes the computed degradation trend vector for the $i$-th battery sample. it is important to note that "Trend" is a high-dimensional feature vector embedded through a deep neural network, encoding the latent feature representation of the battery's current health state and short-term degradation trend. Its length is the same as the degradation trajectory we predict. $\boldsymbol{F}_i$ represents the physics-informed features extracted from the field cycle, and $\boldsymbol{Q}_i$ is the partial charging curve (possibly with random initial SOC) for the field cycle. $N_{expert}$ is the total number of expert networks, and $G_j(\boldsymbol{F}_i)$ is the router

weight associated with Expert$_j$, determined by a degradation-router mechanism. Expert$_j(\mathbf{Q}_i)$ denotes a specific degradation trend estimator under the $j$-th dominant degradation mode.

The FORNN module iteratively integrates AMDP predictions regarding latent degradation trends with assumed future load conditions cycle-by-cycle, enabling proactive degradation trajectory computation without historical data, which can be formulated as:

$$\hat{\mathbf{S}}_i = \text{FORNN}(\text{Trend}_i, \mathbf{Cond}_i) \qquad (2)$$

where, $\hat{\mathbf{S}}_i$ represents the final predicted degradation trajectory (e.g., capacity fade curve) for the $i$-th battery sample over the computation time horizon, $Trend_i$ is the short-term trend output from the AMDP module, and $\mathbf{Cond}_i$ contains future usage conditions such as charge/discharge current, which can be assumed available. The FORNN leverages the sequential modeling capability to capture how future load profiles affect the long-term degradation starting from the current state.

These two modules exhibit unique but complementary strengths that AMDP computes latent degradation trends from learned degradation modes while FORNN deals with second-life use complexities (see Supplementary Fig. 7 for the detailed machine learning pipeline). The degradation routing simultaneously considers physics-informed features reflecting current existing degradation modes and future load profiles to select optimal experts, forming a closed-loop physics-driven specialization and data-driven fusion architecture. Theoretically, iMOE's design philosophy benefits from the success of collaborative learning[1,45], where the weighted collaboration from multiple networks enhances prediction robustness[46–48]. In principle, iMOE applies to degradation trajectory computation across the entire lifecycle stage under arbitrary use conditions, including performance computation in other lifetime stages despite considerably different internal degradation modes.

## Dataset

This work addresses the computation challenge of degradation trajectories for retired batteries under uncertain future use conditions using only partial cycling data at test field, i.e., no historical data is assumed. To validate computation performance when historical and future operating conditions are consistent but historical data is unavailable, we select the Uniform-Life (UL) dataset[49] for demonstration experiment. To validate computation performance when the historical operating conditions are unknown and the batteries are in a deeply aged state, we select the late-stage degradation (LSD) dataset[43] for demonstration experiment. To validate computation performance when the second-life use conditions change significantly as compared to their first-life counterpart, we select the two-phase second-life (TPSL) dataset[23] for demonstration experiment.

The UL dataset comprises batteries tested under different use conditions, with each batch cycled under consistent conditions throughout their entire lifespan until EOL, comprising three batches totaling 130 commercial 18650 cells. Batch 1 consists of $LiNi_{0.86}Co_{0.11}Al_{0.03}O_2$ positive electrode (NCA battery) with 3500 mAh nominal capacity and cutoff voltages of 2.65–4.2 V. Batch 2 contains $LiNi_{0.83}Co_{0.11}Mn_{0.07}O_2$ positive electrode (NCM battery) with 3500 mAh nominal capacity and cutoff voltages of 2.5–4.2 V. Batch 3 includes 42 (3) wt% Li(NiCoMn)O$_2$ blended with 58 (3) wt% Li(NiCoAl)O$_2$ positive electrode (NCM + NCA battery) with 2500 mAh nominal capacity and cutoff voltages of 2.5–4.2 V. All cells underwent cycling in thermal chambers under three temperatures (25 °C, 35 °C, 45 °C) with variable charge rates (from 0.25 C to 4 C) and fixed 1 C discharge rate. Each battery experienced identical operational profiles throughout its full lifecycle until reaching EOL (see Supplementary Table 1 for details). Degradation trajectories in Fig. 2a demonstrate cycling variability.

The LSD dataset contains 86 commercial batteries with $LiNi_{0.5}Co_{0.2}Mn_{0.3}O_2$ cathodes and graphite anodes. These 2.4 Ah batteries (3.7 V nominal) underwent two-phase testing, In Phase 1, the batteries were grouped into 16 distinct charge-discharge protocols to induce diverse degradation behaviors as the state of health (SOH) decreased from 100% to 80% (details in Supplementary Table 2). In Phase 2, all 86 cells were re-cycled from 80% to 50% SOH under a unified low-rate protocol (0.5 C charge/0.2 C discharge) to simulate stationary energy-storage operation. Degradation trajectories in Fig. 2b demonstrate Phase 2 cycling variability. We predict the deep degradation trajectories of second-phase batteries under unknown and varying historical operating conditions.

The TPSL dataset contains batteries subjected to diverse second-life scenarios, where they undergo both randomized and standardized operating conditions following their primary usage phase, containing 77 pieces of 18650 batteries with $LiCoO_2$ and $LiNi_{0.5}Co_{0.2}Mn_{0.3}O_2$ cathodes and graphite anodes. These 2.4 Ah batteries (3.7 V nominal, from 3.0 to 4.2 V cutoffs) underwent two-phase testing: Phase 1 involved 20 cycles of 0.5 C constant-current constant-voltage (CCCV) charging and 2 C discharging. Phase 2 divided batteries into two subgroups: (1) TPSL-Random (55 cells) subjected to stochastic charging profiles with current rates (1 C/2 C/3 C) changed randomly every 5 cycles that follow a uniform statistical distribution, combined with fixed 3 C discharge; (2) TPSL-Fixed (22 cells) cycled under predetermined charge/discharge current combinations (1 C/2 C/3 C) (details in Supplementary Table 3). Figure 2c shows degradation trajectories under uncertain use conditions. In the TPSL-fixed dataset, different magnitudes of charge and discharge currents lead to vastly different degradation trajectories. Meanwhile, in the TPSL-Random dataset, due to variations in the magnitude of the charging current, the difference in the maximum discharge capacity between adjacent cycles can even exceed 0.3 Ah, disproving the assumption of identical operational history for prediction reliability. Figure 2d reveals substantial capacity dispersion at cycles 21, 60, and 100 post-load variation, demonstrating how uncertain loads interact with battery degradations. Figure 2e details the input data construction process, where each sample combines raw charging curve sequences with physics-informed features, see "Methods" (data processing) and Supplementary Note 2.

## Model performance and generalization capability

To validate the robustness of the proposed method under unknown historical data and uncertain second-life use conditions, we conduct evaluations in four real-world application scenarios (UL, LSD, TPSL-Random, TPSL-Fixed). iMOE adaptively computes nonlinear degradation modes under uncertain use conditions through the AMDP module, see Supplementary Table 4 for the associated model parameter.

Performance metrics (RMSE, MAPE, $R^2$) are shown in "Methods" (Evaluation metric). It is emphasized that all experiments use current-cycle data with random initial SOC regions extracted from field conditions to compute degradation trajectories spanning dozens to hundreds of cycles, rather than EOL points (see the prediction horizon selection in Supplementary Note 3). As a time-series computation task, we benchmark the model performance against state-of-the-art (SOTA) sequence computation models[50,51] PatchTST and Informer (see Supplementary Note 4 and Supplementary Table 4 for detailed information). In Fig. 3a, existing research demonstrates that battery degradation can be divided into three typical while distinct degradation phases: SEI formation, SEI thickening, and lithium plating[8,52]. During the battery's lifecycle, the formation and thickening of the SEI layer impacts the battery's capacity degradation in early stages, while lithium-ion deposition typically occurs in the later stages of the battery's use and is one of the main causes of sharp capacity decline. However, it is important to note that the capacity knee at the EOL is not always directly related to lithium plating. In some cases, the capacity

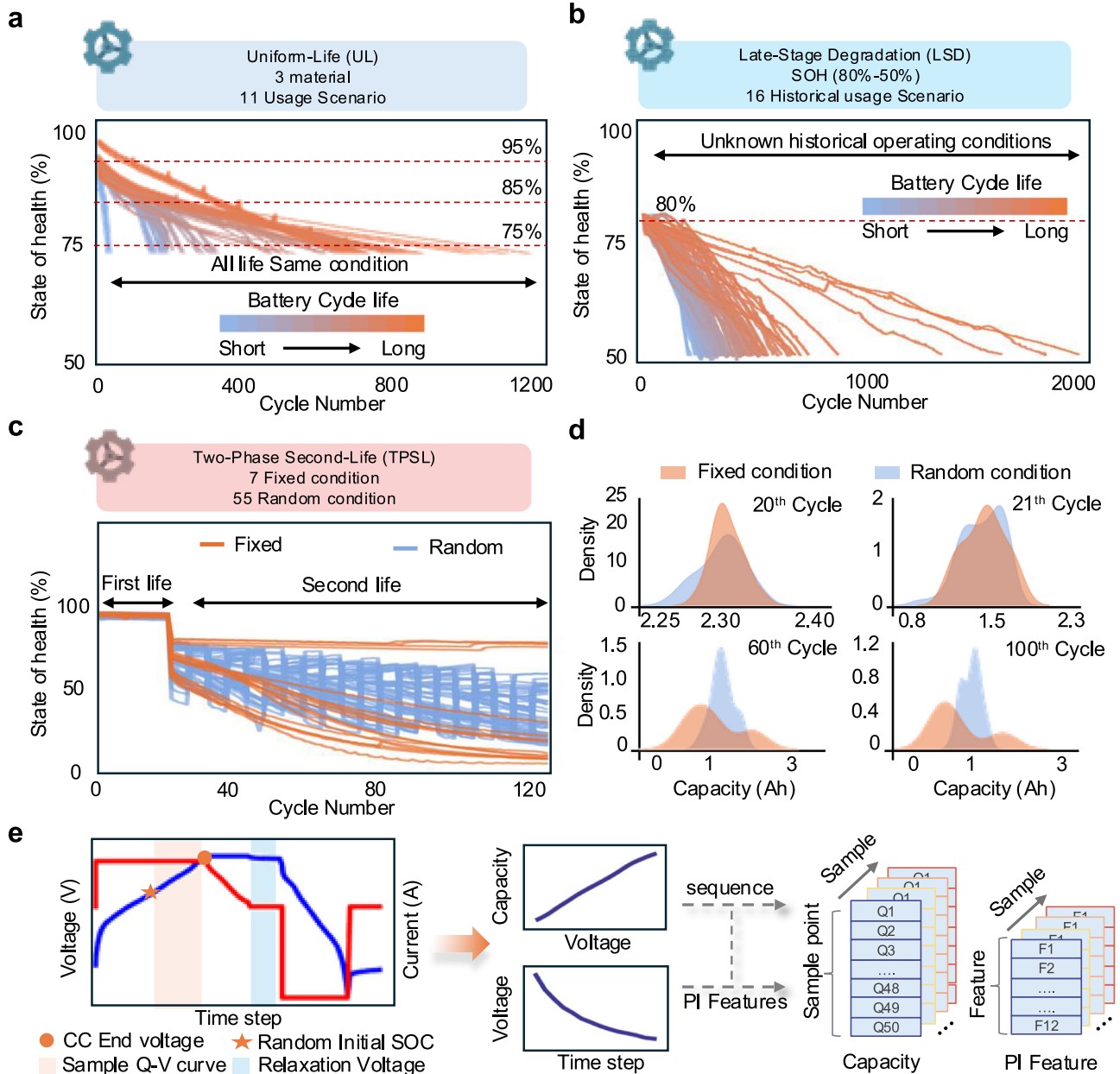

**Fig. 2 | Dataset description. a** Summary of the UL dataset testing conditions and the degradation curves of the batteries. **b** Summary of the LSD dataset testing conditions and the degradation curves of the batteries. **c** Summary of the TPSL dataset testing conditions and the degradation curves of the batteries. **d** Capacities distributions of the cells at different use conditions. **e** The charging curves with random initial state of charge and the 30-min relaxation voltage curves. Source data are provided as a Source data file.

decline at the EOL may also be influenced by other degradation mechanisms, such as SEI thickening, loss of active material, or electrolyte degradation, which can contribute to the sharp capacity drop observed in later stages. The degradation patterns across these phases exhibit significant differences in both available capacity and subsequent aging rates, especially in the slope of the degradation trajectory, with each phase reflecting distinct degradation behaviors[53,54]. Supplementary Fig. 8 illustrates that the definition of degradation stages and mechanisms can be regarded as typical and reasonable. We randomly select representative samples from each degradation stage to analyze performance, where each sample reflects distinct dominant degradation modes with varying capacity retention and aging rates. While PatchTST and Informer show errors across early, middle, and late degradation phases, iMOE demonstrates superior stability, delivering accurate current-cycle capacity estimation while generating

trajectory predictions consistent with phase-specific degradation rates. Supplementary Figs. 9–11 validate iMOE's effective prediction capability across full lifecycle samples.

Figure 3b reveals that in TPSL data, baseline methods display limited capability in capturing long-term trends after future load changes and completely fail during initial load transitions. In contrast, iMOE effectively identified transitional characteristics between phases through future usage condition integration. Supplementary Figs. 12-15 further demonstrate iMOE's successful learning of relationships between future operating conditions and capacity performance. In LSD dataset, the baseline models exhibit larger errors in predicting the later stages of degradation trajectories. This is mainly due to the unknown and varying operating conditions in the first phase, which result in highly heterogeneous degradation behaviors during deep aging. In contrast, our proposed method, benefiting

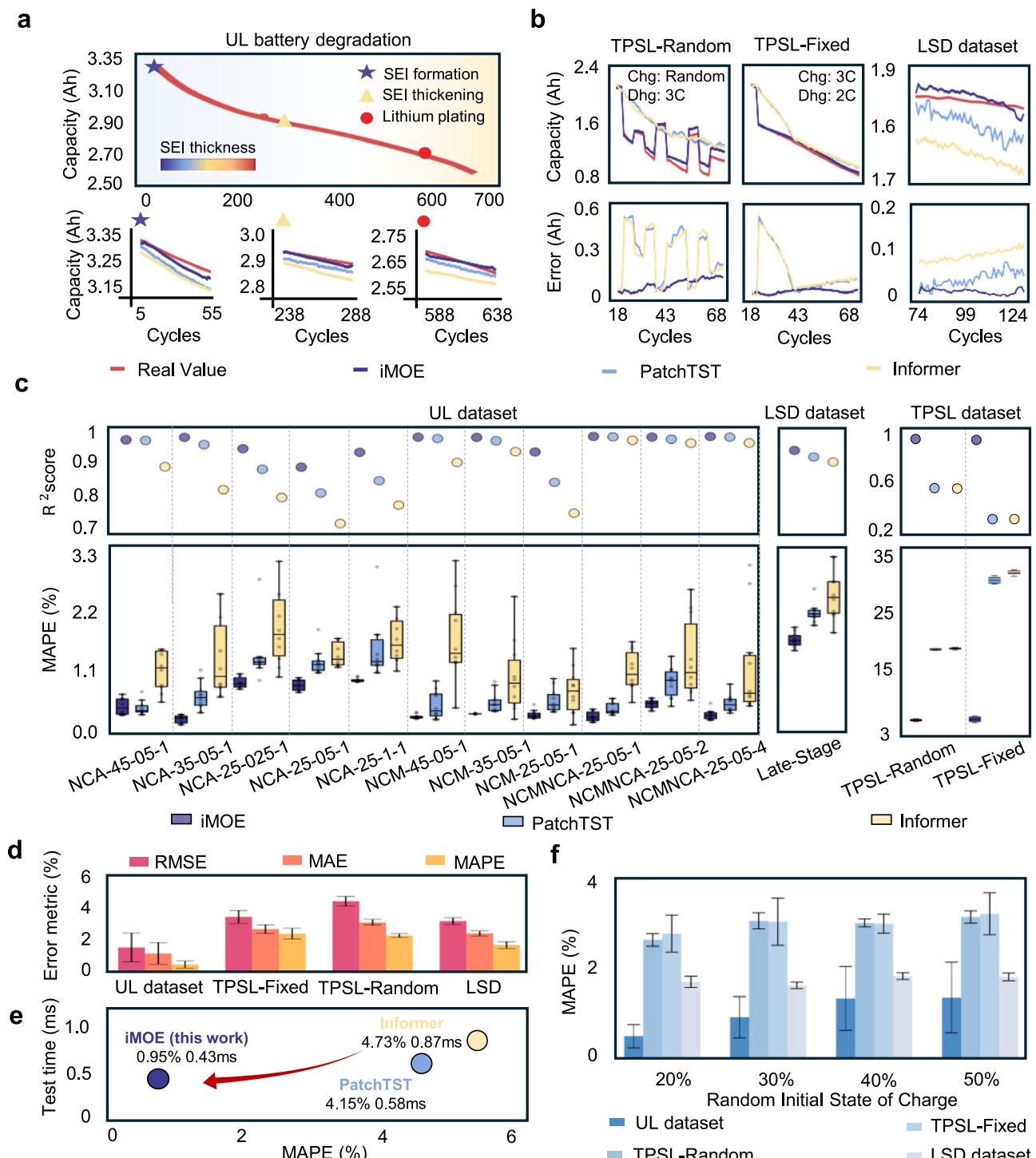

**Fig. 3 | iMOE results and analysis. a** Prediction results of the proposed framework and existing methods for interval samples dominated by three different degradation modes. **b** Prediction results of the proposed framework and existing methods under different secondary use conditions. **c** Comparison of prediction results for different materials and usage conditions. **d** Prediction results for datasets under three different usage conditions. **e** Comparison of prediction time and accuracy between the proposed framework and existing methods. **f** Model performance under different initial SOC conditions in random retirement scenarios. Source data are provided as a Source data file.

from the AMDP module, can achieve more stable and reliable prediction performance.

Figure 3c demonstrates that iMOE exhibits significant and stable advantages in both MAPE and R² metrics with the cross-dataset performance comparison setting. For UL dataset, iMOE achieved its worst prediction result on the NCA-25-1-1 dataset with an average MAPE of 0.98%, while maintaining MAPE below 1.00% across all UL conditions,

showcasing robust prediction performance. PatchTST delivers the second-best results with an average MAPE of 0.85% and a maximum MAPE of 1.55% in the UL dataset, indicating that its patch-based tokenization strategy captures aging variations more effectively than single-point sampling. Informer performs the worst in this scenario, with an average MAPE of 1.48% and a maximum MAPE of 2.13%, suggesting that its sparse attention mechanism may lose critical

information and fail to detect subtle temporal differences in voltage-capacity curves during such extreme prediction tasks. In LSD dataset, iMOE achieves an average MAPE of 1.82% for batteries in the deep degradation stage, whereas the Informer reaches an average MAPE of 2.68%. This result indicates that purely data-driven models struggle to make relatively accurate predictions under the unknown and varying operating conditions of the first-phase usage. In the TPSL dataset, iMOE shows even greater performance advantages over baseline methods, achieving an average RMSE of 0.05 compared to 0.38 for PatchTST and 0.39 for Informer, further confirming the necessity of incorporating future operating conditions in second-life applications.

In Fig. 3d, the scenario analysis confirms that iMOE achieves an average MAPE of 0.52% with a standard deviation of 0.28% for the UL dataset, 1.81% with a standard deviation of 0.13% for the LSD dataset. 2.96% with a standard deviation of 0.44% for the TPSL-Fixed dataset, and 2.81% with a standard deviation of 0.15% for the TPSL-Random dataset. iMOE performed better on the UL dataset with continuous operating history than on the LSD dataset with unknown historical conditions and deep degradation, as well as on the TPSL dataset with highly variable operating conditions (see Supplementary Table 5 for detailed information). This is primarily because continuous operation provides more stable degradation patterns, allowing the model to more accurately capture battery aging behaviors. In contrast, the deep degradation in the LSD dataset exhibits stronger nonlinearity due to differences in aging mechanisms, while the random load switching in the TPSL dataset introduces additional cycle-level nonlinear complexities, increasing the difficulty of degradation trajectory computation. Nevertheless, all MAPE values remained below 3.00%, demonstrating that the proposed model can effectively compute degradation trajectories under uncertain future conditions using only current-cycle data without relying on historical records.

In Fig. 3e, we analyze the relationship between model computational efficiency and performance to meet the real-world deployment requirements. After training, iMOE requires 0.43 ms to infer the entire degradation trajectory in the coming hundreds of cycles incorporating future operating conditions for a single retired battery, while achieving an MAPE of 0.93%. In comparison, baseline methods PatchTST and Informer required 0.58 ms and 0.87 ms per battery, with MAPEs of 4.15% and 4.73%, respectively. iMOE thus achieves the most robust prediction accuracy while emphasizing lightweight real-world deployment.

Unlike existing approaches that rely on fixed voltage or SOC regions and require time-consuming SOC recalibration, we validate iMOE's robustness across five random initial SOC regions. In Fig. 3f, model prediction performance was relatively poorer at an initial SOC of 50%, with MAPEs of 1.44%, 3.35%, and 3.43% for the UL, TPSL-Random, and TPSL-Fixed datasets respectively, accompanied by standard deviations of 0.84%, 0.14%, and 0.50%. Reducing the initial SOC to 20% improved average performance by 63.8%, 16.2%, and 13.7%, respectively.

The observed performance disparity can be attributed not only to the reduced volume of training data available at higher initial SOC from a machine learning perspective but also, more fundamentally, to the inherent lack of adequate Open Circuit Voltage feature information in high-SOC data segments from a physicochemical standpoint, which hinders the model's ability to differentiate degradation modes such as lithium inventory loss and active material loss. Furthermore, as the computations for the UL dataset do not involve cycle-level condition changes, the model outcomes rely more heavily on the data volume at the initial SOC, resulting in more pronounced performance degradation as the initial SOC increases.

## Rationalization of statistical model performance
Here we explore the fundamental mechanisms underlying performance improvements. In Fig. 4a, we first randomly select NCA battery test samples from the UL dataset to visualize the weight assignments of the degradation router throughout their full lifecycle. Without requiring historical data, the degradation router partitions battery lifecycle degradations into three distinct phases using partial cycling data, which is evidenced by existing literature[8,52]. It is observed that early-retirement samples are predominantly governed by Expert Networks 1 and 4, mid-life retired batteries exhibit higher weights for Expert Networks 2 and 3, while late-life retired batteries are primarily regulated by Expert Network 5. These results demonstrate iMOE's capability to identify evolving degradation patterns, with clear functional specialization among expert networks for different degradation stages. Mechanistically, this suggests Expert Networks 1, 3, and 5, respectively, dominate initial SEI layer formation, subsequent thickening, and lithium plating processes, consistent with prior research on primary battery degradation modes[52]. Expert Network 4 maintains high weights during both early and late degradation phases, potentially reflecting its statistical correlation with high degradation rates, i.e., slope of the degradation trajectories, rather than specific mechanistic contributions.

Figure 4b takes Battery 1 as an example to deeply analyze the dynamic correlation between individual cycling samples and expert network weight assignments during capacity degradation. Single-cycle samples at similar degradation stages exhibit clustering characteristics in expert weight allocation, demonstrating highly consistent expert category selection. Meanwhile, the degradation router adaptively adjusts the weights of different expert networks based on the features of individual cycling samples, thereby achieving differentiated prediction of degradation variations both between cycles and across degradation stages. Supplementary Fig. 16 further confirms the observed expert weight dependencies across different battery material systems and second-life use conditions.

We perform T-SNE dimensionality reduction experiments on expert weights across the entire test battery cohort (details in Supplementary Note 5) to visualize the separability of decision boundaries among expert networks. In Fig. 4c, despite significant inter-cell variability, samples from early and late degradation stages form distinct decision boundaries, as demonstrated in Supplementary Figs. 17–19 that iMOE has generalizability across battery material systems and operating conditions. These findings raise a critical question: can the trained iMOE model directly classify retired batteries based on degradation router-generated weight patterns? In essence, this question is an interpretability analysis showing why AMDP should work well.

In Fig. 4d, for the 130 UL dataset batteries retired at different SOH levels (classify solely based on expert weights, see Supplementary Note 6 for criteria), those retired at 95% SOH achieve an "Excellent" classification confidence of 91.5%, while batteries retired at 75% SOH receive a "Degraded" rating of 96%. This further demonstrates that the model utilizes different primary experts for prediction based on the dominant degradation mechanisms at various stages, thereby enabling the reverse classification of retired batteries at different degradation stages according to expert selection. Supplementary Figs. 20–22 and Supplementary Table 6 visualize the expert weight distributions across different SOH, with marked differences further demonstrating our proposed method's novelty and simplicity of AMDP module that can classify the degradation mode using a few electric measurements. Figure 4e examines post-classification computation accuracies of degradation trajectories. iMOE achieves MAPEs of 0.47%, 0.52%, and 0.47% for batteries retired at 95%, 85%, and 75% SOH, respectively. While predictions remain robust for 95%-85% SOH, samples at 75% SOH show maximum outliers, likely due to coupling of accumulated and other unobserved degradation mechanisms.

In Fig. 4f, controlled experiments with noise addition validate feature importance and expert network specialization. By individually applying Gaussian noise ($\sigma = 1$) to each physical feature while holding

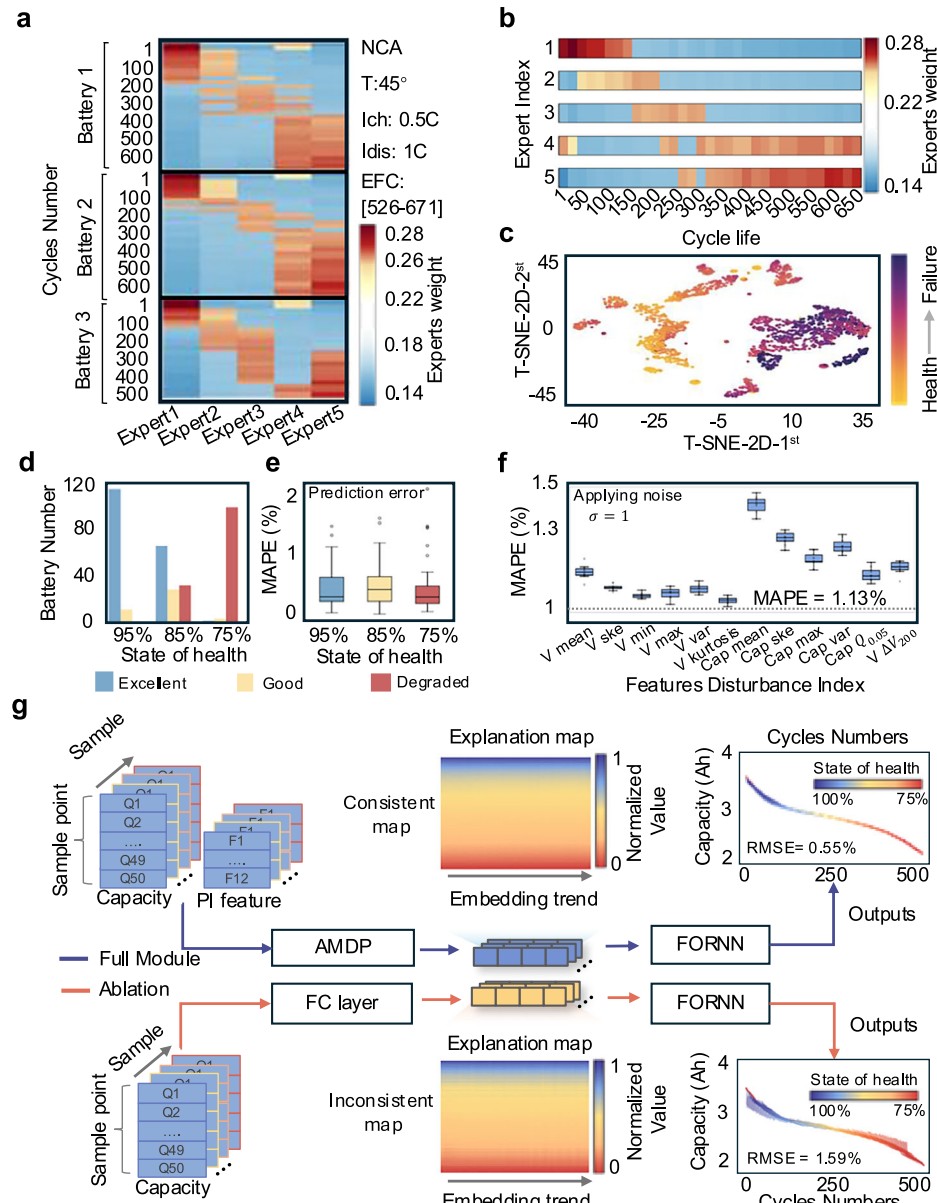

**Fig. 4 | iMOE interpretability. a** Visualization of degradation expert weights for selected NCA batteries. **b** Degradation expert weights corresponding to different intervals in random retirement scenarios throughout battery lifecycle. **c** Visualization of expert weights after T-SNE dimensionality reduction to 2D. **d** Classification results of UL dataset batteries under different SOH retirement scenarios using expert weights only (specific classification scheme see Supplementary Note 6). **e** Prediction results for batteries retired at different SOH levels. **f** Prediction results in noise injections with varying physical features. **g** Internal architecture and signal analysis diagram of the iMOE. Source data are provided as a Source data file.

other features at their original value, we observe elevated errors from baseline MAPE of 1.13% when features are perturbed with noise. Supplementary Figs. 23–24 present the performance analysis under conditions of missing values and reduced feature inputs. With the same amount of noise injections, the feature whose perturbation causes the largest decrease in degradation trajectory computation accuracy is regarded as the most important. It reveals that features extracted from relaxation voltage that is reflective of internal resistance demonstrate greater noise resilience than those from capacity-voltage curves (particularly their mean values, after adding noise, the MAPE increased from 1.13% to 1.38%, representing a 22% relative increase, as these means directly reflect charge storage capacity per voltage increment). The observed performance decrease caused by misrouting samples to expert networks not specialized for their specific degradation phases (e.g., late-stage samples erroneously routed to Expert Network 1

instead of 5) confirms the functional divergence among expert networks, enabling history-free computation of degradation mode. Supplementary Figs. 25–26 show reduced specialization when degradation trajectories are geometrically similar, while demonstrating that iMOE's superior performance does not strictly rely on any single physical feature. Supplementary Fig. 27 and Supplementary Note 7 presents the SHAP and Spearman importance analysis for different features. We further analyze the model results after removing features with lower SHAP importance in Supplementary Fig. 28. The results indicate that, in certain scenarios, redundant features can be removed to maintain a balance between computational efficiency and accuracy.

In Fig. 4g, we investigate the contribution of the AMDP module to predict accuracy using an ablation experiment. When replacing AMDP with a simple linear layer while keeping other components unchanged, comparative analysis of the embedded representations

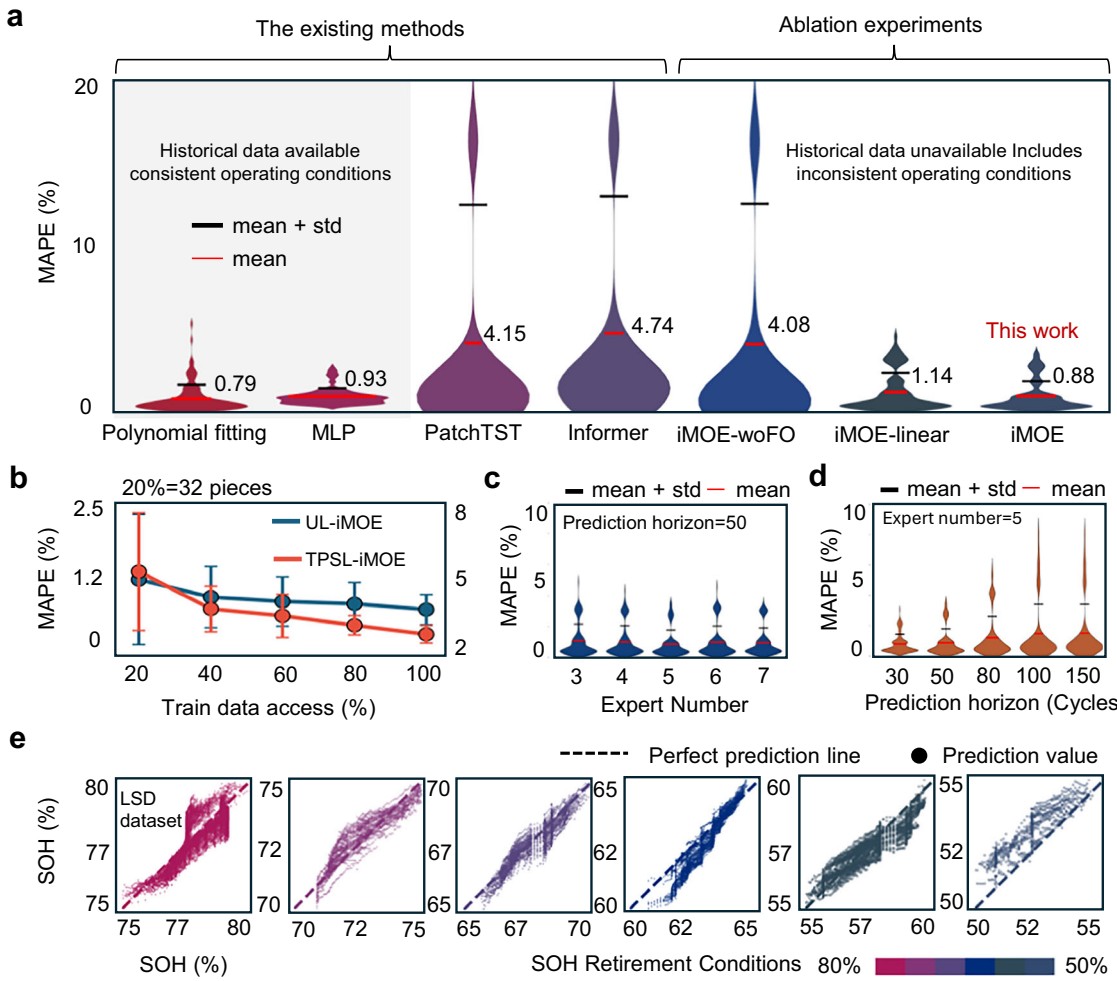

**Fig. 5 | Uncertainty analysis. a** Comparison of MAPE error distributions for degradation trajectory predictions (gray areas represent conditions with available historical data and consistent secondary use conditions; blank areas represent comparisons with other models and ablation study results). **b** Model accuracy under limited training data conditions. Impact of (**c**) expert quantity and (**d**) prediction horizon on model accuracy. **e** Prediction results across different SOH ranges. Source data are provided as a Source data file.

(see Supplementary Note 8 for details) shows that while the ablated model can roughly distinguish degradation stages, it fails to capture subtle SOH variations within similar stages. This results in discontinuous embeddings and unreasonable trajectory predictions, highlighting AMDP's critical role in enabling robust long-term predictions under history-free settings, Supplementary Fig. 29 directly shows the magnitudes of different cycling trend metrics. By decoding implicit semantics of expert weights, iMOE transforms from a prognostic tool into an interpretable decision engine, bypassing traditional sequential assessment workflows that require extensive data curation and model calibration in real-world deployment.

## Uncertainty analysis

Under the assumption of available historical data and consistent second-life operational conditions, we extend our analysis to scenarios where partial historical data is accessible. While this significantly simplifies the prediction task, it does not align with the complex data composition of real-world applications, using polynomial fitting and MLP-based approaches to map historical trajectories to future degradation (see implementation details in Supplementary Note 9). In Fig. 5a, without requiring battery-specific mechanistic analysis, a simple polynomial regression and single-layer MLP achieved a 0.79% and a 0.93% MAPE, respectively. We demonstrate the iMOE's capability in computing degradation trajectories using historical data regarding maximum capacity to validate its adaptability to data-accessible scenarios (see Supplementary Fig. 30 and Supplementary Note 10 for implementation details), With 10 historical maximum capacity points accessible, iMOE achieves an average MAPE of 0.53% when predicting the next 50 cycles. We stress that these historically-dependent methods become infeasible when historical data is unavailable in real-world settings, underscoring the practical limitations of such assumptions. To quantify component contributions of AMDP and FORNN, we designed two ablation studies: (1) replacing AMDP with a linear layer (iMOE-linear); and (2) substituting FORNN with a standard RNN (iMOE-woFO). After replacing the AMDP architecture with a linear layer directly, the overall MAPE increased from 0.95% to 1.19%, which aligns with the observation in Fig. 4e that the embedding vectors became indistinguishable between different degraded samples after removing the AMDP architecture. This demonstrates that the adaptive integration of predictions from different degradation modes contributes to robust and high-precision estimation. When replacing FORNN with a standard RNN, the model exhibited a significant decline in prediction performance for TPSL dataset. This further underscore the necessity of incorporating future load conditions in second-life scenarios, as future operating conditions often differ from those in their first-life service patterns.

Figure 5b evaluates the iMOE performance under limited training data access (data partitioning details are provided in Supplementary Note 11). When training data was reduced from 100% to 40%, the model

performance maintained stable on UL dataset, achieving an average MAPE of 0.75% with a 0.56% standard deviation. However, a linear performance decrease was observed on TPSL dataset, with an average MAPE increasing to 3.86% with a 0.88% standard deviation. When training data was reduced to 20% (32 batteries in total, pruned training data size 5MB), both datasets exhibited noticeable error and uncertainty increase, with MAPE reaching 1.07% for UL and 5.30% for TPSL. This training data access experiment indicates that learning future operating conditions' impact on battery degradation trajectories requires higher data curation costs, whereas the UL dataset demonstrates relatively lower data access dependency. Despite this observed data access challenge, Supplementary Tables 7–9 compare baseline model performance under scarce data access, validating iMOE's superior generalization capability in few-shot learning scenarios.

The decomposition of degradation subspaces and dominant mechanism prediction depend on the number of experts and TopK selection (see "Methods"). Figure 5c examines the impact of number of experts on the iMOE performance. The iMOE demonstrates relative insensitivity to hyperparameter modifications, across experiments with five distinct expert network quantities, the average MAPE was 0.98%. when employing 4 experts, the model showed comparatively poorer performance with an average MAPE of 1.03%, while the 6-expert configuration exhibited inferior stability with a standard deviation of 1.04%. After carefully balancing practical deployment requirements against performance metrics, we ultimately adopted a configuration with 5 experts and TopK = 2.

Different application scenarios have varying requirements for prediction horizons. In practical applications, it is desirable to predict degradation trajectories as far as possible while maintaining acceptable performance. As shown in Fig. 5d, we present results across different prediction lengths. When the prediction length is reduced, model performance improves, achieving an average MAPE of 0.80%. In most cases, model performance degrades with increasing prediction length. Nevertheless, even when predicting 150 future cycles from current-cycle data, the average MAPE remains below 1.50%, demonstrating iMOE's robustness in long-term forecasting. Supplementary Tables 10–12 compare our method's performance with baseline approaches under extended prediction horizons. The results show that all models exhibit performance degradation as prediction length increases. When predicting future 100-cycle degradation trajectories, PatchTST and Informer achieve average MAPEs of 8% and 7.98%, respectively, while iMOE maintains a lower average MAPE of 1.47%, enabling more accurate long-term predictions. We further analyzed the model's predictive performance under unknown and varying historical operating conditions at different levels of deep degradation. As shown in Fig. 5e, across six SOH intervals from 80% to 50%, the model achieved consistently good prediction accuracy, with an average MAPE of 1.81% and a standard deviation of 0.16%. However, when SOH fell within the 55–50% range, the prediction accuracy decreased, with an average MAPE of 2.00%, reflecting the increased difficulty of modeling degradation trajectories when multiple degradation mechanisms interact at advanced aging stages.

## Discussion

Existing research paradigm of degradation trajectory prediction predominantly relies on historical data access and assumed identical future use conditions, which is often not true for second-life batteries[1]. This paradigm inflicts either expensive lifecycle data acquisition or time-consuming capacity calibration tests post-retirement, also raising safety concerns due to its inability to account for second-life use uncertainty. The proposed iMOE network, after offline training, utilizes data already available in battery management systems in deployment scenarios to directly predict degradation trajectories under assumed but uncertain future second-life use conditions, without relying on historical data, eliminating the need for extra offline testing. The success of iMOE stems

from adaptively and interpretably classifying degradation modes, statistically the geometric slope of the degradation trajectory at the observation point, to inform use-condition-dependent degradation trajectory prediction in the extended second-life use.

Through case studies involving four usage scenarios (i.e., UL, LSD,TPSL-Random, TPSL-Fixed), 295 independent cells, and 93 secondary-use operating conditions, we demonstrate that iMOE achieves a promising lifecycle MAPE of 0.95% in computing degradation trajectories of 50 future cycles using only real-time online data, while reducing computation time by up to 50% compared to the SOTA time-serials computational models. The average MAPE values across all validated datasets were reduced by 77% as compared to PatchTST and Informer. It is argued that iMOE is flexible with fixed and random SOC regions, allowing for seamless integration into second-life batteries that exhibit random initial SOC distributions at the collection field. By adaptively integrating degradation mode classification using proposed AMDP module and assumed uncertain future use conditions, the battery degradation trajectory prediction horizon can be extended to 150 cycles, with an average MAPE of 1.50% with a maximum MAPE below 6.26%. Hyperparameter sensitivity analysis on the number of experts confirms model's robustness under uncertain second-life use conditions, with an average MAPE of 0.98%. The interpretability of the iMOE is demonstrated via the visualization of latent embeddings from expert weights that represent degradation modes, thus the model is physically interpretable.

iMOE model shows great potential to transform the battery reusing and recycling industry landscape by reducing the need of manual-assisted testing approach to an automated data-driven decision support system. However, it must be acknowledged that iMOE approach primarily relies on electrochemical characteristics (such as voltage-capacity relationships and relaxation voltage) for macroscopic diagnostics. While extensive validation confirms the physical understanding of these features[19,49], they essentially remain statistical correlations instead of physical causality. It is recommended that future work should focus on non-invasive in-situ sensing signals[33,34,55], such as vibration sensing, strain sensing, ultrasonic signals, fiber optic sensing to enhance internal state observation for a physics-informed modeling, instead of physically interpretable presented in this work. Building a "physics-statistics" fusion framework has the potential to improve the interpretability of iMOE[36]. For example, parameters from battery electrochemical or equivalent circuit models could be updated online as an intermediate physical interpretation layer for degradation trajectory prediction. Integrating primary governing equations of battery modeling with data-driven methods would also provide physical interpretability[39].

With acknowledging the assumption that future use conditions can be assumed, it is reasonable for verifying the effectiveness of iMOE under predefined uncertain scenarios with considerable uncertainties, specifically, consistent operating histories, deeply aged batteries with unknown prior use and varying second-life conditions with 95 use conditions being validated. However, despite practical applications such as home energy storage and integrated photovoltaic-storage systems often have pre-defined operational plans, the actual operating conditions of battery energy storage systems remain even more uncertain and are difficult to predict[56,57]. Although this work has discussed the importance of introducing future use conditions in the FORNN structure, the analysis of the uncertainty of future use conditions needs to be extended to highly random and time-varying use scenarios. Thus, future work could focus on introducing load conditions at finer temporal resolutions to ensure the validity of predicted use conditions of retired batteries. Another aspect is that random initial SOC is an important factor in the data availability of retired batteries. Although the effectiveness of iMOE in predicting degradation trajectory under different initial SOC conditions has been validated, the initial SOC of retired batteries is more uncontrollable due to the time-consuming

nature of acquisition such information. Future work suggests considering SOC normalization during the feature extraction phase or introducing an SOC-aware routing mechanism in the model, to better handle battery degradation difference under different SOC states[58]. Although the paper has considered four very different application conditions, namely UL, LSD, TPSL-Random, TPSL-fixed, given that the second-life application of the battery is safety-critical, more extreme conditions need to be investigated. This is because the robustness of the dataset to non-steady-state use conditions, such as the battery with thermal runaway or internal short circuits, has not been included. Considering the bias of the training data is crucial, and it is necessary to assess the uncertainty and confidence intervals of the predicted degradation trajectories, as UL and TPSL data hardly cover all environmental fluctuations under actual service conditions, such as thermal non-uniformity at the battery pack level. Future work is suggested to collect real-world variables that include different manufacturers, degradation mechanisms, and cell-to-cell variations.

This work demonstrates the potential of history-free, proactive degradation trajectory computation amid second-life use uncertainty, reducing the need for time-consuming and costly offline testing while ensuring safety compliance. The iMOE has demonstrated technical feasibility in retired battery deployment decision-making at scale by laying groundwork for light, interpretable and generalizable battery management models. Broadly, this work underscores the promise of integrating physics insights into predictive health management models for early malfunction detection and reliability assessment, paving the way for safer and more durable operation of critical infrastructures such as energy storage systems.

## Methods

### Data preparation
**Input signals.** We consider two signals in each cycle: partial charging profile that starts at a random voltage (i.e., a random initial SOC) and ends at the cutoff voltage. Relaxation voltage curve recorded immediately after charging. Let $I(t)$, $V(t)$, and $t$ denote the instantaneous current, voltage, and time, respectively. We track the battery voltage from $V_{min}$, which varies across samples due to random SOC, up to the cutoff voltage $V_{end}$.

**Partial charging curve.** We divide voltage range $[V_{min}, V_{end}]$ into $M$ increments of size $\Delta V$. The partial charging curve $q$ denotes the cumulative input charge as a function of voltage:

$$q = [q_0(V), q_1(V), \cdots, q_M(V)], \quad q_i(V) = \int_{t(V_{min})}^{t(V_{min} + i\Delta V)} |I(t)| dt \quad (3)$$

In experiments, we record 50 segments at equal intervals (i.e., $M = 50$), yielding a 50-dimensional sequence.

**Relaxation voltage curve.** To capture the relaxation behavior, we measure the open-circuit voltage $v_k(t)$ for $k = 0, 1, ..., M$ at 30-minute intervals after charging concludes:

$$v = [V_0(t) \, V_1(t) \cdots V_M(t)] \quad (4)$$

This sequence of length $M + 1$ reflects how the polarized voltage recovers over time.

**Feature engineering.** From partial charging and relaxation voltage signals, we extract 12 features derived from physical considerations, including polarization overcharge, relaxation slope, and voltage plateau transitions. All 12 features are normalized to [0, 1] before entering the model. The final input to the iMOE consists of the partial charging curve $q \in \mathbb{R}^{50}$ and the feature vector $F_i \in \mathbb{R}^{12}$ derived from both the charging and relaxation profiles.

## Model architecture

The proposed interpretable mixture of deep expertized learning network consists of two modules, *Adaptive Multi-degradation Prediction* (AMDP) module and a *Future-Operation Recurrent Neural Network* (FORNN), to include current-cycle degradation phenomena and prospective operating conditions that may arise in second-life applications.

### Adaptive multi-degradation prediction (AMDP)
Formally, let $F_i \in \mathbb{R}^{12}$ be physics-informed features for the $i$-th sample, and $q = [q_0(V), q_1(V), \cdots, q_M(V)]$ be its partial charging curve. The AMDP model defines the expert networks, each denoted Expert$_j$, $j = 1, 2, \ldots, E$. A degradation mode routing $F_i \mapsto g$ produces a probability weight $g(F_i)$ across these experts:

$$g(F_i) = Softmax(\text{TopK}(Softmax(\text{H}(F_i)), k)) \quad (5)$$

where, $H(\mathbf{F})\mathbb{R}^E$ is a feed-forward transformation with a noise term:

$$\text{H}(F_i) = F_i W_g + \psi \, \text{Softplus}(F_i W_{noise}) \quad (6)$$

where, $W_g \in \mathbb{R}^{12 \times E}$ and $W_{noise} \in \mathbb{R}^{12 \times E}$ are learned parameters, and $\psi \in N(0, 1)$ is drawn from a standard Gaussian. The top-$k$ experts are selected and normalized by a $Softmax$ function.

Retired batteries may undergo multiple degradation mechanisms, e.g., SEI formation, SEI thickening, loss of active material, and lithium plating, that can coexist or interact. To flexibly and specifically model these different pathways, AMDP module enables a MOE structure by fusing predictions from multiple expert submodels, each "specialized" in a certain degradation mode. Within the AMDP enabled MOE framework, an Expert is not confined to a single degradation mode; instead, it is represented as a weighted superposition of countably many implicitly defined "typical degradation mode". Let the $j$-th Expert$_j$ be denoted as:

$$\text{Expert}_j(q_i) = \sum_{\theta \in \Omega} w_i(\theta) \phi(q_i; \theta) \quad (7)$$

where $q_i$ represents the partial charging sequence extracted under a random initial SOC condition, $\phi(q_i; \theta)$ is the basis response associated with the latent degradation mode indexed by $\theta$, and $w_i(\theta)$ is the weight that Expert$_j$ assigns to corresponding degradation mode. The output is a short-term (within a small region around the computation point) degradation trend under the assumption that the $j$-th mechanism is dominant. The final AMDP output for sample $i$ is a weighted sum:

$$Trend_i = \sum_{j=1}^{E} g(F_i)\text{Expert}_j(q_i) : \mathbb{R}^N \to \mathbb{R}^L \quad (8)$$

Where $Trend_i \in \mathbb{R}^{1 \times L}$ represents the preliminary degradation trend output by AMDP module, and $L$ indicates length of the predicted future degradation trajectory. Expert$_j$ denotes the predictor for a specific degradation mode, where $g_j(F_i)$ is the weight for expert $j$. This step produces a latent degradation trend vector $Trend_i$ for subsequent cycles, reflecting the dominant mechanism(s) underpinned by $F_i$. Hence, the model adaptively fuses multiple degradation mode subspaces into a short-term trend embedding for subsequent long-horizon computation.

### Future-operation recurrent neural network (FORNN)
Second-life applications involve future load conditions that can alter degradation pace as well as the maximum available capacity in the current cycle. To include these influences, we employ a recurrent neural network that takes as input AMDP, as output $Trend_i$ and a set of

future load parameters:

$$C_i = \left[ \left( I^{(1)}_{\text{charge}}, I^{(1)}_{\text{discharge}}, T^{(1)} \right), \ldots, \left( I^{(L)}_{\text{charge}}, I^{(L)}_{\text{discharge}}, T^{(L)} \right) \right] \quad (9)$$

where $C_i \in \mathbb{R}^{L \times 3}$, $I^{(\ell)}_{\text{charge}}$, $I^{(\ell)}_{\text{discharge}}$, and $T^{(\ell)}$ represent the charge current, discharge current, and temperature in future cycle $\ell$. For each future load condition, its length is equal to L, and the length of the degradation trajectory needs to be predicted. We concatenate these load vectors with the short-term trend from AMDP:

$$\boldsymbol{X_i} = \text{Concat}(Trend_i, C_i) \quad (10)$$

The LSTM processes $X_i \in \mathbb{R}^{L \times 4}$ sequentially to predict the capacity trajectory $\hat{S}_i$ over $L$ future cycles:

$$\hat{S}_i = \text{LSTM}(\boldsymbol{X_i}) \quad (11)$$

By iterating cycle by cycle, FORNN accommodates a prediction horizon that the user chooses, merging the degradation modes signature from AMDP with the load sequence the battery will experience under the second-life complexities and uncertainties.

## Model training

We train the AMDP and FORNN modules jointly by minimizing a loss function that balances trajectory accuracy with degradation router diversity. Let $S_i$ be the ground-truth capacity trajectory of length $L$ for sample $i$, the true capacity label represents the maximum discharge capacity of that cycle. Let $\hat{S}_i$ be the corresponding model output. The trajectory fidelity objective is

$$\ell_{\text{traj}} = \frac{1}{N} \sum_{i=1}^{N} ||\hat{S}_i - S_i||^2 \quad (12)$$

where the summation is over the training set of size $N$.

Because MOE can exhibit "expert collapse" (where one expert is used for all samples), we include a regularization term to encourage usage across experts. For a mini-batch, the router weight assigned to expert $j$ is averaged as

$$A_j = \sum_{i \in \text{Batch}} g_j(\boldsymbol{F_i}) \quad (13)$$

We encourage these averages to avoid large variation by penalizing their coefficient of variation:

$$\ell_{router} = \frac{\text{Var}(A_1, A_2, \ldots, A_E)}{\left(\text{Mean}(A_1, A_2, \ldots, A_E)\right)^2 + \varepsilon} \quad (14)$$

where $\varepsilon$ is a constant set to 10, Var and Mean are the variance and mean operator, respectively. The total loss is:

$$\ell = \alpha \, \ell_{\text{traj}} + \beta \, \ell_{router} \quad (15)$$

with $\alpha$ and $\beta$ controlling the trade-off between trajectory accuracy and expert diversity. We set $\alpha = 0.75$ and $\beta = 0.25$ in our experiments.

## Evaluation metrics
### Root mean square error (RMSE)

$$\text{RMSE} = \sqrt{\frac{1}{N} \sum_{i=1}^{N} \left(\hat{S}_i - S_i\right)^2} \quad (16)$$

## Mean absolute percentage error (MAPE)

$$\text{MAPE} = \frac{1}{N} \sum_{i=1}^{N} \frac{\left| \hat{S}_i - S_i \right|}{S_i} \times 100\% \quad (17)$$

## Coefficient of determination ($R^2$)

$$R^2 = 1 - \frac{\sum_{i=1}^{N} \left(\hat{S}_i - S_i\right)^2}{\sum_{i=1}^{N} \left(S_i - \bar{S}\right)^2} \quad (18)$$

where, $N$ is the total number of samples (i.e., cycles), $S_i$ and $\hat{S}_i$ are the true and computed capacity, respectively. $\bar{S}$ is the sample mean of the ground-truth capacities in the computation horizon.

## Data availability
Raw data can be found here[23,43,49]. Source data are provided as a Source Data file. Source data are provided with this paper.

## Code availability
The code used in this work has been deposited at the Zenodo repository here[59].

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

## Acknowledgements

This work was funded by the National Natural Science Foundation of China (Grant No. 51877120) (B.X.), the Key Scientific Research Support Project of the Shanxi Energy Internet Research Institute (Grant No. SXEI2023A002) (X.Z.), the Meituan Scholar Program-International Collaboration Project (Grant No. 202209A) (X.Z.), the Tsinghua Shenzhen International Graduate School-Shenzhen Pengrui Young Faculty Program of Shenzhen Pengrui Foundation (Grant No. SZPR2023007) (G.Z.), the Guangdong Basic and Applied Basic Research Foundation (Grant No. 2023B1515120099) (G.Z.), and the National Natural Science Foundation of China (No. 92572103).

## Author contributions

Xinghao Huang: Writing—review and editing, writing—original draft, visualization, validation, methodology, formal analysis, conceptualization. Shengyu Tao: Writing—review and editing, writing—original draft, visualization, validation, methodology, formal analysis, conceptualization, supervision, project administration. Chen Liang: Writing—review and editing, visualization. Yining Tang: Writing—review and editing, visualization. Jiawei Chen: Writing—review and editing, methodology, formal analysis. Junzhe Shi: Writing—review and editing, methodology, formal analysis. yuqi li: writing—review and editing, methodology, formal analysis. Bizhong Xia: Resources, supervision, project administration, funding acquisition, conceptualization. Guangmin Zhou: Supervision, project administration, writing—review and editing, supervision, conceptualization. Xuan Zhang: Supervision, project administration, writing—review and editing, supervision, conceptualization.

## Funding

## Competing interests

The authors declare no competing interests.

## Additional information

[1]Tsinghua Shenzhen International Graduate School, Tsinghua University, Shenzhen, China. [2]Department of Civil and Environmental Engineering, UC Berkeley, Berkeley, CA, USA. [3]Department of Electrical Engineering, Chalmers University of Technology, Gothenburg, Sweden. [4]Civil and Environmental Engineering, Stanford University, Stanford, CA, USA. [5]Institute for Artificial Intelligence, Peking University, Beijing, China. [6]Department of Materials Science and Engineering, Stanford University, Stanford, CA, USA. [7]Center of International Innovation for Technology and Science, Shenzhen, Guangdong, China. [8]These authors contributed equally: Xinghao Huang, Shengyu Tao. ✉e-mail: shengyu.tao@chalmers.se; xiabz@sz.tsinghua.edu.cn; guangminzhou@sz.tsinghua.edu.cn; xuanzhang@sz.tsinghua.edu.cn

