## [Transparent Peer Review file · Nature Communications]

iMOE: Prediction of second-life battery degradation trajectory using interpretable mixture of experts

Corresponding Author: Professor Shengyu Tao

Version 0:

Reviewer comments:

Reviewer #1

(Remarks to the Author)

Dear Author,

Thanks for sharing your research work. This reviewer has carefully considered the contents, quality, and extent, providing the following feedback.

The state-of-the-art analysis is weak. The use of historical data and some examples of existing works are discussed, but not strong enough to formulate the research questions.

Similarly, the proposed PIMOE, AMDP, and FORNNs are not justified, not enough existing works are discussed, and no motivation is available behind choosing this path.

What is this 3rd type of battery mentioned as NMC+NCA? What is the cathode composition?

In Supplementary Figure 8a/b, there are some red lines referring to low SoH that existed since the beginning. How is it possible?

Why does Supplementary Figure 10 show spikes appear after every 100 cycles, probably referring to rest periods, while others do not exhibit such?

Supplementary figures 11-14 show time-to-time capacity recovery, which is strange. Could the authors explain this?

There are many figures presented in the supplementary document, but this reviewer does not find the significance. Rather, they could be grouped/tailed or less sampled/categorized. It is more important to state the findings from these curves. Could the authors motivate the choice behind?

There is confusion about the availability of the selected parameters from noisy, missing, or partial curves. It is unclear in the manuscript if not absent.

Many acronyms are not explained at their first use. Please consider the documents separately.

Supplementary tables should be placed at the beginning of the supplementary file.

The UL and TPSL datasets are referred to literature, meaning the PIMOE framework is the only novel contribution in this research. The work seems to have weaker novelty. Could the authors comment on this or guide to the novel contributions?

Why PatchTST and Informer have bad predictions with TPSL datasets?

In the main manuscript, in the early section of page 23, the authors tried to link the aging mechanisms to the expert networks. What is the validity of such a hypothesis?

The research is regarding PI-physics informed parameters, but there is no discussion of these parameters on the impact of aging and how they are relevant to be called physical parameters. They are driven by electrical tests and more statistical data.

What is the justification for Figure 4d, where the SEI and Li-plating are plotted separately, while these events could happen in the same cell, depending on the activated mechanisms (vs conditions).

(Remarks on code availability)

Reviewer #2

(Remarks to the Author)

This is an interesting paper addressing the challenge of predicting battery degradation trajectories in second-life applications using a novel physics-informed mixture of experts (PIMOE) architecture. The manuscript is generally well-

written, and the proposed approach is innovative, combining domain knowledge with machine learning for interpretable predictions. The results are compelling and relevant to both academia and industry.

However, while the paper shows promise, there are several important issues that must be addressed before it can be considered for publication.

1) The approach assumes that future load conditions are known in advance, which is a strong assumption in real-world second-life scenarios. Although this is briefly acknowledged in the discussion, a more thorough examination of the limitations and implications of this assumption is warranted. Furthermore, scenarios involving uncertain or stochastic future use should be considered or at least simulated.

2) While the model is claimed to be "physics-informed," its interpretability largely relies on correlations extracted from capacity-voltage and relaxation curves. These features, while derived from empirical electrochemical knowledge, do not truly integrate physics-based modeling such as differential equations or parameter estimation from equivalent circuit models. A clearer distinction between physics-informed and statistically correlated is needed.

3) The paper compares its approach to two baselines (Informer and PatchTST), which are relevant time-series models. However, additional baselines from battery-specific modeling domains—such as hybrid physics-informed neural networks or direct state-of-health estimation models—would strengthen the empirical evidence. Also, it is unclear whether the baselines were retrained under the same conditions, which affects reproducibility.

4) The following two recent and highly relevant works in the field of imitation learning-based degradation modeling are not cited and must be referenced:

• Pozzi, A., Incremona, A. & Toti, D. "Imitation learning-driven approximation of stochastic control models". *Applied Intelligence*, 55, 838 (2025). <https://doi.org/10.1007/s10489-025-06704-x>

• Pozzi, A., Incremona, A., Toti, D. "Neural Network-Based Imitation Learning for Approximating Stochastic Battery Management Systems," *IEEE Access*, vol. 13, pp. 71041-71052, 2025. <https://doi.org/10.1109/ACCESS.2025.3563300>

These works are directly relevant to the idea of modeling stochastic system dynamics using data-driven expert systems and should be included for context and completeness.

5) The current model provides deterministic predictions. In safety-critical applications like battery reuse, it is essential to assess the uncertainty or confidence bounds of the predicted degradation trajectories. The paper mentions this briefly in the discussion but does not quantify it in the results. A demonstration or discussion of uncertainty-aware predictions (e.g., via ensembles or Bayesian methods) is encouraged.

(Remarks on code availability)

Reviewer #3

(Remarks to the Author)

This is a well-written paper on a general topic of interest to the battery community. I can recommend publication if the authors address the following comments.

1) Spearman coefficient & SHAP values

The authors rely on their physics-informed router + experts structure, but it would strengthen the paper to validate feature importance explicitly:

- A simple Spearman correlation analysis between input features and outputs (SOH, degradation modes) could confirm monotonic relationships and highlight redundancies among the 12 features.
- A SHAP (Shapley Additive Explanations) analysis on the trained neural model could provide insight into which features the network relies on most. This would help connect the "physics-informed" explanation with the actual feature usage inside the NN.
- Suggestion: Add a supplementary analysis of feature importance (Spearman + SHAP) to improve interpretability and confidence in the model.

2) Reduction in Number of features

The manuscript states that the input consists of 12 features derived from partial charge + relaxation voltage profiles. While 12 is not excessive for modern ML, some may be correlated or redundant.

- Suggestion: Discuss whether feature selection or dimensionality reduction (e.g., PCA, correlation filtering, or SHAP pruning) could simplify the model without significant loss in accuracy. This also makes the physics-informed routing more interpretable.

3) Effect of random SOC starting point

The work does not discuss how the initial SOC (random starting point of partial charging) influences the extracted features. In practice, this can strongly affect the relaxation trajectory and hence the prediction accuracy.

- Ignoring this factor may lead to optimistic performance compared to real-world deployment, where SOC is not always controlled.
- Suggestion: Even if beyond the current study's scope, add a short discussion of this limitation and how future work might address it (e.g., SOC normalization, SOC-aware routers). This would show awareness of practical challenges.

4) Aging trajectory

One of the key assumptions in the developed model is that the aging trajectory can be divided into three distinct zones: SEI formation, SEI thickening, and lithium plating. However, this assumption lacks sufficient rationale or supporting evidence for the aging datasets analyzed. The authors should elaborate on this degradation classification, ideally providing experimental

or model-based justification. Furthermore, it would be helpful if the authors discuss the sensitivity of their methodology/result to the choice of degradation mechanism. For instance, how would the approach change if loss of active material were considered as one of the dominant degradation pathways?

5) Dataset

The authors should make it clearer that the aging datasets used in this work are obtained from the literature and provide the corresponding references.

6) Typos and grammatical mistakes

- Line 56: relavent → relevant
- Line 65: Larbor → labor
- Line 65: awared → aware
- Line 79: the demanding → demanding
- Line 95: implication → implication
- Line 123: "uncertain secondary use condition" → "uncertain secondary use conditions"
- Line 279-280: "a interpretability" → "an interpretability"
- Line 389: exhitbit → exhibit
- Line 399: extenstive → extensive

(Remarks on code availability)

Reviewer #4

(Remarks to the Author)

Authors describe a method for estimation of Li-ion cell capacity and combine it with predictive model that allows forecasting of future capacity degradation trends. Authors claim that the proposed method allows history-free estimation of State-of-Health (SOH) of the battery hence eliminating a need for historical usage records or extensive lab tests. Elimination of data requirements is a very valuable technical goal since unavailability of historical usage or test data is a biggest obstacle for accurate estimation of SOH.

While authors claim history-free method, the training process used for training of the estimation and prediction components of the proposed model relies on extensive test data collected offline. I think that authors need to revise some of their claims related to this discrepancy. That training data is the missing 'history'. It is very well known that with sufficiently rich degradation data collected offline one can train a data-driven model to achieve pretty much any desired accuracy. I recommend the authors to put a bit more effort in connecting the proposed modeling method to physics of Li cell degradation. Without this connection between selected ML features and vaguely mentioned degradation modes the paper does not stand out from a long list of data-driven modeling papers for SOH estimation.

Some more technical comments:

- Line 130: Please, explain what the Trend vector is exactly. Is it a vector of capacity or SOH values over one cycle vs time? Several cycles?

- It would be very valuable to a reader to see an explanation of why/how selected AMDP features contain information about SOH of the cell. Without this info a reader stays uneducated about why the proposed method should work and paper reads just like yet another 'we stacked several MLMs to achieve good KPIs' paper.

- Figure 2c and subsequent figures show fluctuations in measured capacity by double-digit percent. How is this possible? How was exactly capacity measured in experiments?

- There is a large drop in capacity from First to Second life on Figure 2c. Why is there such a drop?

- Figure 3b, and 2c show not only drops but also jumps up to 20% in measured capacity. Can you, please provide explanation of what is going on there? This is not physical behavior for NCM/NCA cells.

- Line 199-200. Capacity knee at EOL of a cell, 'third phase', is not always related to Li-plating.

- Line 207. Again, not clear why measured capacity jumps around and why the model follows this trend.

- Line 243. I think the conclusion about use of higher SOC range is a bit ML model development centric. A physical reason could be that the snippet of data starting at SOC > 50% simply does not contain enough information about OCV features that can help the model to distinguish between components of degradation such Li loss vs AML.

- Figure 3: What is the prediction horizon? Are we predicting x-number of cycles ahead or the entire second life until certain SOH?

- Line 375, Discussion: PIMOE needs reach experimental offline test data to train on according to the described methodology. Please state that explicitly.

- Line 461, equation 5: What is exactly the 'basis response' in equation 5? What are exactly the degradation modes?

- I think if degradation modes are described that may help to explain 'physics-informed' part of the title. Without this explanation I do not see any 'physical' components in the proposed model.

- Equation 6: what are the units of 'Trend'? Is it in Ah or normalized SOH value? Is it a vector of values vs time or delta V? L is the number of samples in a cycle or many cycles?

- Equation 7: Future load parameters, I(t), T(t). Are these time vectors of dimension N?

- Equation 10: How are true capacity labels generated?

Noticed typos:

- Line 89, 'interatively'

- Line 100, 'extends'

- Line 170, 'pecices'

- Line 388, 'valuse'

(Remarks on code availability)

Version 1:

Reviewer comments:

Reviewer #1

(Remarks to the Author)

In Supplementary Table 1, please check the material composition typos. The NMC composition is just copied and pasted from NCA.

Thanks for adding the LSD dataset to the analysis.

Regarding the linked aging mechanisms, the identified/mentioned mechanisms (i.e. SEI thickness growth) should be described using words like 'may be' or 'probably' as they could not be validated within the scope, although they are proven in other works. The batteries are always different from one to another and trigger different levels of mechanisms under known circumstances.

(Remarks on code availability)

The code seems clean but did not run it to verify.

Reviewer #2

(Remarks to the Author)

I appreciate the thorough revision. All my previous concerns have been fully addressed, and I have no further comments.

(Remarks on code availability)

The code package appears to be well organized and adequately documented. It includes a clear README. I was able to run the code without particular difficulties, and the results align with those reported in the manuscript.

Reviewer #3

(Remarks to the Author)

The authors have delicately revised the paper based on my comments and I am satisfied with the revised version. As such, I recommend the publication of this work.

(Remarks on code availability)

Reviewer #4

(Remarks to the Author)

I would like to thank the authors for putting the effort in addressing reviewer's questions and suggestions. The manuscript has improved, and most important technical aspects of the proposed method are now more readable and easier to understand. I think the results are valuable and should be published.

My main concern is still about the claim of physical interpretability claimed in the title and abstract. From my point of view the proposed method does not really provide any physical interpretability of the results. Weight assignments by PIMOE experts are a result of ML model training. The model simply learns to differentiate between different capacity degradation trends provided in the training data. The model uses its available flexibility to put more weight on one part of sub-model closer to BOL and more weight on another part of sub-model later in cells' life. There is nothing physical about it, it is simply a result of a properly setup optimization problem. Fig 30 in supplemental material and its notes does not really tell a reader about physical changes in the cells: it does not show how much Li or active materials have been lost or how kinetic parameters of the cells have changed. The way it is presented, it is just a collection of curves that ML model learned to map to target values.

I suggest keeping the references to physical interpretations in the main text and supplemental materials as is, but to modify the title and remove the corresponding claim from the abstract and the introduction. It will be fairer to a reader and avoid any confusion.

Supplementary note 5 refers to figure 4e. Could not find anything related to Trend on Fig 4e of the article.

Tables 1,2,3 in supplementary material pdf are not properly formatted, difficult to read.

(Remarks on code availability)

Responses to Reviewer Comments

Title	Physics-informed mixture of experts network for interpretable battery degradation trajectory computation amid second-life complexities
Revised Title	PIMOE: Physically interpretable mixture of experts network for battery degradation trajectory prediction amid second-life complexities
Authors	Xinghao Huang 1, Shengyu Tao 1, 2, 3, *, Chen Liang 1, Jiawei Chen 5, Junzhe Shi 2, Yuqi Li 6, Bizhong Xia 1, *, Guangmin Zhou1, *, Xuan Zhang 1, *
Revised Authors	Xinghao Huang 1, Shengyu Tao 1, 2, 3, *, Chen Liang 1, Yining Tang4, Jiawei Chen 5, Junzhe Shi 2, Yuqi Li 6, Bizhong Xia 1, *, Guangmin Zhou1, *, Xuan Zhang 1, 7 *
Journal	Nature Communications
Manuscript ID	NCOMMS-25-53313

Table of Contents

Response to Reviewers.....	4
Response to Reviewer #1	4
Response to Comment 1	4
Response to Comment 2	7
Response to Comment 3	10
Response to Comment 4	11
Response to Comment 5	13
Response to Comment 6	14
Response to Comment 7	15
Response to Comment 8	15
Response to Comment 9	17
Response to Comment 10	17
Response to Comment 11	17
Response to Comment 12	23
Response to Comment 13	24
Response to Comment 14	28
Response to Comment 15	30
Response to Reviewer #2	32
Response to Comment 1	32
Response to Comment 2	39
Response to Comment 3	46
Response to Comment 4	47
Response to Comment 5	48
Response to Reviewer #3	50
Response to Comment 1	50
Response to Comment 2	52
Response to Comment 3	54
Response to Comment 4	57
Response to Comment 5	61
Response to Comment 6	61

Response to Reviewer #4	62
Response to Comment 1	62
Response to Comment 2	70
Response to Comment 3	76
Response to Comment 4	77
Response to Comment 5	81
Response to Comment 6	81
Response to Comment 7	82
Response to Comment 8	83
Response to Comment 9	83
Response to Comment 10	84
Response to Comment 11	85
Response to Comment 12	86
Response to Comment 13	87
Response to Comment 14	88
Response to Comment 15	92
Response to Comment 16	93
Response to Comment 17	94
Response to Comment 18	95
References	96

**Response to Reviewers**

**Response to Reviewer #1**

Dear Reviewer,

The authors respond to your suggestions point by point and carefully revise the manuscript and
supplementary information, where you can find corresponding revisions using “*track changes*”
mode. Changes are colored in *blue* in revised clean version. The authors truly hope that the responses
appropriately address your justified concerns, and that revised manuscript meets your expectations.

**Comment 1**

The state-of-the-art analysis is weak. The use of historical data and some examples of existing works
are discussed, but not strong enough to formulate the research questions.

**Response to Comment 1**

Thank you for the comment. The authors have revised the paper in the following aspects:

● The authors elaborated on existing methods and literature, providing a detailed discussion of
the limitations of current studies that rely on historical data and fixed future operating
conditions for predicting degradation trajectories of retired batteries.

● The authors provided a detailed outlook on future work in the Discussion section.

The authors have made modifications in the introduction.

[revised manuscript text omitted]

**Comment 2**

Similarly, the proposed PIMOIE, AMDP, and FORNNs are not justified, not enough existing works
are discussed, and no motivation is available behind choosing this path.

**Response to Comment 2**

Thank you for the comment. The authors have revised the paper in the following aspects:

- ● The PIMOE framework includes two core modules, AMDP and FORNN. The authors have
revised the Introduction section to emphasize the importance of physical interpretability and
necessity of incorporating future operating conditions, providing a more comprehensive
explanation of our motivation.

[revised manuscript text omitted]

Comment 3

What is this 3rd type of battery mentioned as NMC+NCA? What is the cathode composition?

Response to Comment 3

Thank you for the comment. In this paper, "NCM+NCA" refers to a hybrid cathode material system

composed of 42 (±3) wt.% Li(NiCoMn)O₂ blended with 58 (±3) wt.% Li(NiCoAl)O₂, with graphite

used as the anode material. The authors have revised the paper in the following aspects:

- The authors have provided a more detailed explanation of the battery material composition and characteristics in the main text

- The authors have revised Supplementary Tables 1 and 2 avoid potential ambiguity.

We have made the following modifications to the article content:

“The UL dataset comprises batteries tested under different use conditions, with each batch cycled

under consistent conditions throughout their entire lifespan until EOL, comprising three batches

totaling 130 commercial 18650 cells. Batch 1 consists of LiNi_{0.86}Co_{0.11}Al_{0.03}O₂ positive electrode

(NCA battery) with 3500mAh nominal capacity and cutoff voltages of 2.65–4.2 V. Batch 2 contains

LiNi_{0.83}Co_{0.11}Mn_{0.07}O₂ positive electrode (NCM battery) with 3500mAh nominal capacity and cutoff

voltages of 2.5–4.2 V. Batch 3 includes 42 (3) wt.% Li(NiCoMn)O₂ blended with 58 (3) wt.%

Li(NiCoAl)O₂ positive electrode (NCM + NCA battery) with 2500mAh nominal capacity and cutoff

voltages of 2.5–4.2 V. All cells underwent cycling in thermal chambers under three temperatures

(25 °C, 35 °C, 45 °C) with variable charge rates (from 0.25C to 4C) and fixed 1C discharge rate.

Each battery experienced identical operational profiles throughout its full lifecycle until reaching

EOL (see Supplementary Table 1 for details). Degradation trajectories in Fig. 2a demonstrate

cycling variability, while Supplementary Fig. 28 illustrates significant remaining useful life (RUL)

differences among batteries.

”

Modifications have been made in the Supplementary Table 1 and 2:

Cell types	Material	Working conditions			Nominal capacity (Ah)	Cut-off voltage (V)	Number of cells
		Charge rate	Discharge rate	Temperature (°C)			
NCA	Li _{0.86} Ni _{0.86} Co _{0.11} Al _{0.03} O ₂	0.5	1	25	3.5	2.65-	66

	3 O2 / Graphite	0.5	1	35		4.2	
		0.5	1	45			
		1	1	25			
		0.25	1	25			
	42 (3) wt.%	0.5	1	25			
NCM+NC	Li(NiCoMn)O2 blended	0.5	2	25	2.5	2.5–	9
A	with 58 (3) wt.%	0.5	4	25		4.2	
	Li(NiCoAl)O2/ Graphit						
		0.5	1	25			
NCM	Li0.86Ni0.86Co0.11Al0.0	0.5	1	35	3.5	2.5-4.2	55
	3 O2 / Graphite	0.5	1	45			

Cell types	Material	Working conditions			Nominal capacity (Ah)	Cut-off voltage (V)	Number of cells
		Charge C rate	Discharge C rate	Temperature (° C)			
		2	1	25			3
		3	1	25			3
		1	2	25			4
	LiCoO2 and	2	2	25			3
NCM	LiNi0.5Co0.2Mn0.3O2 /	3	2	25	2.4	3.0-4.2	3
	graphite anodes	2	3	25			3
		3	3	25			3
		Random current	3	25			55
		(1C~3C)					

Comment 4

In Supplementary Figure 8a/b, there are some red lines referring to low SoH that existed since the beginning. How is it possible?

Response to Comment 4

Thank you for the comment. Some of the red lines indicate discrepancies in the early stage of battery degradation, reflecting relatively large deviations between predicted and true values during the initial phase—rather than implying low initial state of health in the battery's label. The authors have revised the paper in the following aspects:

- The authors have revised all the figures to ensure clear and unambiguous presentation.

Modifications have been made in the Supplementary Figure 8-10:

“Predicting the degradation trajectory of arbitrary NCM material battery samples throughout their full lifecycle. Using a sliding window approach for data extraction and prediction (see

*Supplementary Note 1 for details), the method predicts the next 50 cycles for individual batteries*
 *based solely on partial data from the current cycle. (a)-(c) present prediction results under different*
 *operating conditions. The solid black line represents the true SOH values, while the color scheme*
 *denotes the predicted values for the health states of different samples used as input. (Naming*
 *conventions are specified in Supplementary Table 1)."*

**Comment 5**

Why does Supplementary Figure 10 show spikes appear after every 100 cycles, probably referring
 to rest periods, while others do not exhibit such?

**Response to Comment 5**

Thank you for the comment. Battery with NCM+NCA materials underwent additional
 electrochemical impedance spectroscopy tests during the capacity degradation experiment. The tests
 were conducted at full charge state within a frequency range of 10 kHz to 0.01 Hz with a sinusoidal
 amplitude of 250 mA. Additionally, a 60-minute open-circuit voltage rest period was implemented
 prior to each impedance test. The rest of the period and testing procedures resulted in a slight
 capacity recovery of the battery. The authors retained this cycle in the experiment as such transient
 capacity recovery phenomena are commonly observed in real-world scenarios, even though
 excluding it might have further improved prediction accuracy.

The authors have revised the paper in the following aspects:

- ● The authors made the following modifications to the Supplementary Table 1:

The authors have made the following modifications to the Supplementary Table 1:

*“The table presents the material types, operating conditions, nominal capacities, cutoff voltages,*

*and number of batteries used in our experimental dataset. All batteries in this study underwent*
*identical load cycling throughout their full lifecycle until aging. The dataset is designated as the UL*
*Dataset³³, and we adhere to the naming convention established by the data providers. For instance,*
*the notation "NCA-45-05-1" refers to an NCA battery tested at 45°C with 0.5C CCCV charging and*
*1C discharging conditions, with similar naming logic applying to other cases. For the NCM + NCA*
*battery, the electrochemical impedance is conducted every 50 cycles at full charge in a range of 10*
*kHz to 0.01 Hz (6 data points per decade of frequency) with a sinusoidal amplitude of 250 mA. 60*
*min are set at the open circuit voltage before the electrochemical impedance tests. The UL Dataset*
*encompasses 3 material types, 11 operating conditions, and 130 batteries in total. Notably, for each*
*sample, we utilize only the current cycle's data without requiring any historical cycle information. ”*

**Comment 6**

Supplementary figures 11-14 show time-to-time capacity recovery, which is strange. Could the
authors explain this?

**Response to Comment 6**

Thank you for the comment. The TPSL dataset incorporates 56 types of randomized charge-
discharge protocols to simulate diverse secondary-use conditions with high complexity and
stochasticity. These varying protocols result in significant differences in the maximum available
capacity even for the same battery across consecutive cycles. This inherent variability invalidates
methods that assume consistent future operating conditions (e.g., classical time-series models). In
real-world secondary applications of retired batteries, future load profiles are often unpredictable,
further highlighting the complexity of our task. The authors acknowledge that without further
clarification, this aspect may cause confusion among readers.

The authors have revised the paper in the following aspects:

- ● The authors made the following modifications to the article content:
- ● The authors have made the following modifications to the Supplementary Table 2:

The authors have made the following modifications to the article content:

*“Fig. 2c shows degradation trajectories under uncertain use conditions. In the TPSL-fixed dataset,*
*different magnitudes of charge and discharge currents lead to vastly different degradation*
*trajectories. Meanwhile, in the TPSL-Random dataset, due to variations in the magnitude of the*

*charging current, the difference in the maximum discharge capacity between adjacent cycles can*
*even exceed 0.3 Ah, disproving the assumption of identical operational history for prediction*
*reliability.”*

The authors have made the following modifications to the Supplementary Table 2:

*“The table presents the materials, operating conditions, nominal capacities, cutoff voltages, and*
*number of batteries used in our experimental dataset. After the initial 20 cycles, the load conditions*
*were modified for these batteries, with 22 batteries undergoing different constant operating*
*condition cycles and 55 batteries subjected to random operating condition cycles. The dataset is*
*designated as the TPSL Dataset¹⁶.*

*In this study, "TPSL-Random" refers to batteries operating under random conditions during their*
*second-life usage, while "TPSL-Fixed" indicates batteries operating under fixed conditions during*
*their second-life usage. The UL Dataset encompasses a total of 66 second-life operating conditions*
*(55 random and 11 fixed) across 77 batteries. In the TPSL-Fixed dataset, batteries under different*
*fixed conditions exhibited significantly divergent degradation trajectories. In the TPSL-Random*
*dataset, the random operating conditions resulted in notable maximum capacity differences even*
*between adjacent cycles. This further underscores the necessity of incorporating future operating*
*conditions as conditional input. Notably, for each sample, we utilize only the current cycle's data*
*without requiring any historical cycle information. “*

**Comment 7**

There are many figures presented in the supplementary document, but this reviewer does not find
the significance. Rather, they could be grouped/tabled or less sampled/categorized. It is more
important to state the findings from these curves. Could the authors motivate the choice behind?

**Response to Comment 7**

Thank you for the comment. The authors have revised the paper in the following aspects:

- ● some figures have been grouped and presented in table format.

**Comment 8**

There is confusion about the availability of the selected parameters from noisy, missing, or partial
curves. It is unclear in the manuscript if not absent.

**Response to Comment 8**

Thank you for the comment. The authors acknowledge that in reality, noisy, incomplete, or partially
missing curves are very common and represent an urgent issue that needs to be considered.

Therefore, in Supplementary Figure 27, The authors have already evaluated the robustness of the
 proposed method by adding noise to the extracted physical features. However, your suggestion is
 crucial evaluating only the physical feature with added noise is insufficient.

The authors have revised the paper in the following aspects:

- ● The authors have made the following modifications to Supplementary Figure 23 to evaluate
 the model performance under conditions of incomplete intervals and noisy data.

The authors have made the following modifications to the Supplementary Note 2:

*“ Supplementary Figure 23. Analysis of the Impact of Sensor Noise and Missing Values on Results*
 *In real-world applications, sensor sampling errors are inevitable. This section aims to investigate*
 *the impact of sensor sampling errors on estimation results. Specifically, we applied three levels of*
 *random Gaussian noise to partial charging curves and relaxation voltages extracted from the field.*
 *As shown in Figure (a), different colors represent varying magnitudes of Gaussian noise. On the*
 *selected dataset, as the sampling error increases, the model's error also increases linearly.*
 *In extreme cases, incomplete data sampling or missing values may occur. We also considered the*
 *impact of such scenarios on the results. Specifically, we randomly removed 20%, 40% of the*
 *sampling points, with missing values handled using simple linear interpolation. As shown in Figure*
 *(b), different colors represent different levels of missing data. On the selected dataset, as the amount*
 *of missing data increases, the model's error also rises. “*

Comment 9

Many acronyms are not explained at their first use. Please consider the documents separately.

**Response to Comment 9**

Thank you for your comment. The authors have revised the paper in the following aspects:

- ● The authors have double-checked and revised the explanations of acronyms upon their first
appearance.

**Comment 10**

Supplementary tables should be placed at the beginning of the supplementary file.

**Response to Comment 10**

Thank you for the comment. The authors have placed all supplementary tables at the beginning of
the supplementary file to ensure a more intuitive presentation of the supplementary file.

**Comment 11**

The UL and TPSL datasets are referred to literature, meaning the PIMOIE framework is the only
novel contribution in this research. The work seems to have weaker novelty. Could the authors
comment on this or guide to the novel contributions?

**Response to Comment 11**

Thank you for the comment. Although the UL and TPSL datasets are indeed sourced from existing
literature, we would like to emphasize that the core innovation of the PIMOIE framework lies in its
physics-interpretable MOE architecture for “history-free” degradation trajectory prediction of
second-life batteries. This method enables the interpretable calculation of degradation trajectories
for retired batteries in deployment scenarios using only partially available field signals, without
requiring complete historical data.

The authors have revised the paper in the following aspects:

- ● The authors further analyzed a newly released retired battery dataset to more comprehensively
evaluate the superiority of the proposed method under different second-life scenarios.
- ● The authors have added descriptions of the newly introduced dataset and revised Figure 2 to
provide more comprehensive coverage of retired battery scenarios.
- ● The authors have included performance analyses of the new dataset and modified Figure 3 to
evaluate the prediction performance of deeply degraded batteries
- ● The authors have added performance evaluations across different SOH intervals in the
uncertainty analysis section.
- ● The authors have added detailed information of the new dataset in Supplementary Table 3.

The authors further analyzed a newly released retired battery dataset to more comprehensively
 evaluate the superiority of the proposed method under different second-life scenarios. The authors
 have added the following description to the main text:

“ The LSD dataset contains 86 commercial batteries with $\text{LiNi}_{0.5}\text{Co}_{0.2}\text{Mn}_{0.3}\text{O}_2$ cathodes and
 graphite anodes, These 2.4 Ah batteries (3.7 V nominal) underwent two-phase testing. In Phase 1,
 the batteries were grouped into 16 distinct charge–discharge protocols to induce diverse
 degradation behaviors as the SOH decreased from 100% to 80% (details in Supplementary Table
 3). In Phase 2, all 86 cells were re-cycled from 80% to 50% SOH under a unified low-rate protocol
 (0.5 C charge / 0.2 C discharge) to simulate stationary energy-storage operation. Degradation
 trajectories in Fig. 2b demonstrate Phase 2 cycling variability, we predict the deep degradation
 trajectories of second-phase batteries under unknown and varying historical operating conditions. ”

**Fig. 2 Dataset Description.** (a) Summary of the UL dataset testing conditions and the degradation

curves of the batteries. (b) Summary of the LSD dataset testing conditions and the degradation

*curves of the batteries. (c) Summary of the TPSL dataset testing conditions and the degradation*
*curves of the batteries. (d) Capacities distribution of the cells at different use conditions. (e) The*
*charging curves with random initial state of charge and the 30-minute relaxation voltage curves.*

The authors have added the following description in the Model performance and generalization
capability:

*“Fig. 3b reveals that in TPSL data, baseline methods displayed limited capability in capturing*
*long-term trends after future load changes and completely failed during initial load transitions. In*
*contrast, PIMOE effectively identified transitional characteristics between phases through future*
*usage condition integration. Supplementary Figs.11-14 further demonstrate PIMOE's successful*
*learning of relationships between future operating conditions and capacity performance. In the LSD*
*dataset, the baseline models exhibited larger errors in predicting the later stages of degradation*
*trajectories. This is mainly due to the unknown and varying operating conditions in the first phase,*
*which resulted in highly heterogeneous degradation behaviors during deep aging. In contrast, our*
*proposed method, benefiting from the AMDP module, achieved more stable and reliable prediction*
*performance.*

[revised manuscript text omitted]

The authors have added the following description in the Supplementary Table 2:

**“Supplementary Table 2. Technical Specifications of LSD Dataset.**

The table presents the materials, operating conditions, nominal capacities, cutoff voltages, and
number of batteries used in our experimental dataset. During phase I, as the SOH of the batteries
declined from 100% to 80%, 86 cells were divided into 16 groups, with each group subjected to a
distinct charge–discharge protocol. In phase II, as the SOH further decreased from 80% to 50%, all
86 cells were cycled under a unified protocol. The dataset is designated as the LSD Dataset.

In the phase 2 dataset, although all batteries were operated under identical cycling conditions,
differences in the loading conditions during their first-life usage led to varying impacts on internal
degradation mechanisms, resulting in significantly divergent degradation trajectories during the

*second-life phase. This phenomenon highlights the profound influence of historical usage conditions*
 *on the long-term health evolution of batteries and provides an ideal validation scenario for*
 *evaluating the model’s predictive capability under unknown historical conditions and deep*
 *degradation states. We utilized only the second-phase dataset to simulate the model’s ability to*
 *predict degradation trajectories of deeply aged, retired batteries in deployment scenarios where*
 *historical data are unavailable.*

Cell types	Material	Working conditions			Nominal capacity (Ah)	Cut-off voltage (V)	Number of cells
		Charge C rate	Discharge C rate	Temperature (°C)			
Phase 1	LiCoO ₂ and LiNi _{0.5} Co _{0.2} Mn _{0.3} O ₂ / graphite anodes	0.5	1	25	2.4	3.0-4.2	8
		0.5	1	25			8
		0.5	1	25			6
		1	1	25			8
		1.5	1	25			8
		2	1	25			4
		0.5	2	25			4
		0.5	3	25			4
		1	1	25			4
		1	1	25			4
		1.5	1	25			4
		1.5	1	25			4
		2	3	25			4
		0.5	0.2	25			3
		0.5	0.5	25			7
0.5	2	25	4				
Phase 2		0.5	1	25			86

**Comment 12**
 Why PatchTST and Informer have bad predictions with TPSL datasets?

Response to Comment 12

Thank you for the comment. The TPSL dataset incorporates 66 types of randomized charge-discharge protocols to simulate diverse secondary-use conditions characterized by high complexity and stochasticity. Under such variable operational profiles, existing methods including classical time series models like PatchTST and Informer—often assume future operating conditions remain

consistent with historical data. Although capable of modeling battery temporal degradation, these
approaches fail to adapt to changing future operational patterns, leading to suboptimal prediction
performance. The fundamental limitation lies in their inability to effectively capture the correlation
between the future maximum available capacity of second-life batteries and the uncertainty of future
usage conditions.

The authors have made the following modifications to the article content:

*“Fig. 3b reveals that in the TPSL data, baseline methods exhibited limited capability in capturing*
*long-term trends following future load changes and failed completely during the initial load*
*transition phase. This is because models like PatchTST and Informer, as classical time-series models,*
*predict the future by mining temporal patterns from historical data. However, in the TSPL dataset,*
*the uncertain charge and discharge currents significantly impacts the maximum available capacity*
*of the current cycle. In contrast, PIMOE effectively identified transitional characteristics between*
*phases by integrating future usage conditions. Supplementary Figs. 11-14 further demonstrate*
*PIMOE’s successful learning of the relationships between future operating conditions and capacity*
*performance.*

**Comment 13**

In the main manuscript, in the early section of page 23, the authors tried to link the aging
mechanisms to the expert networks. What is the validity of such a hypothesis?

**Response to Comment 13**

Thank you for the comment. In Figure 4a, a clear correlation can be observed between battery
samples at different degradation stages and the corresponding expert weights. In Figure 4b and
Supplementary Figures 19–21, the authors further demonstrate the relationship between aging
mechanisms and expert networks more directly through t-SNE clustering. In Supplementary Figures
16–18, samples with different SOH values show distinct expert network selections. However, a
deeper connection between the underlying degradation mechanisms and the expert networks still
needs to be established.

The authors have revised the paper in the following aspects:

- ● The authors further illustrate the correlation between degradation mechanisms at different
stages of degradation and the expert network by analyzing the IC curves of the dataset.

● The authors further validated the rationality of these physical features through SHAP analysis
and Spearman correlation coefficients.

The authors further illustrate the correlation between degradation mechanisms at different stages of
degradation and the expert network by analyzing the IC curves of the dataset.

*“ Supplementary Figure 28. IC Aging Curve Analysis of the Dataset*

*This section aims to demonstrate the rationality and universality of dividing battery aging into*
*multiple stages using IC curves, and to explain how these parameters reveal the battery degradation*
*mechanisms and reflect their physical significance. This assumption is based on numerous studies*
*on lithium battery degradation patterns. For instance, existing research suggests that during the*
*battery's lifecycle, the formation and thickening of the SEI layer affect the capacity degradation in*
*the early stages, while lithium-ion deposition typically occurs in the later stages of battery use and*
*is one of the main causes of the sharp capacity decline.*

*IC curves reveal internal changes in the battery by showing the subtle variations between battery*
*capacity and voltage, especially during the aging process, where the SEI layer growth, lithium-ion*
*loss, and internal electrochemical reactions significantly impact the IC curve. As shown in the figure,*
*different colors represent different SOH samples. In the early charging and discharging cycles, the*
*curve displays sharp peaks, indicating the battery is in a healthy state with significant and rapid*
*capacity changes, mainly reflecting the formation of the SEI layer and surface chemical reactions.*
*As the cycle count increases, the peaks of the IC curve become more rounded and shift to the right,*
*and longer flat sections appear, indicating that the battery's reversible capacity is gradually*
*decreasing. When the battery enters a more severe degradation phase, the peaks of the IC curve*
*gradually decrease. At this point, degradation mechanisms such as lithium-ion deposition and loss*
*of active materials dominate the capacity decline. The widening of the peaks in the curve and the*
*sharp capacity drop indicate that the battery's energy storage capacity is severely compromised.*

In Supplementary Figures 1-6, we directly display the correlation between these physical features
 and battery aging. Furthermore, we have further validated the rationality of these physical features
 through SHAP analysis and Spearman correlation coefficients.

We have made the following modifications to the Supplementary Figure 22:

*“ To quantify the influence of input features on the model's predictions and elucidate the*
 *underlying physical mechanisms, we employed SHapley Additive exPlanations (SHAP) analysis and*
 *Spearman's rank correlation analysis. As shown in Fig. a, the SHAP analysis based on NCA+NCM*
 *battery data reveals the differential contributions of the 12 physics-informed features to the final*
 *prediction outcome. Concurrently, Spearman's correlation analysis in Fig. b demonstrates that most*
 *features exhibit statistically significant correlations with SOH. These two analytical results*
 *corroborate each other, collectively confirming the robustness and reliability of the selected physical*
 *features in effectively characterizing the battery degradation state.*

Based on the strong correlation between the selected input features of the AMDP module and SOH,
 the degradation gating network can effectively distinguish data samples from different degradation
 stages to make corresponding predictions. By directly performing t-SNE clustering on the weights
 of different experts, PIMOE shows clear clustering boundaries for samples from different
 degradation stages, further demonstrating the effectiveness of the MoE architecture in identifying
 different degradation stages.

*“Supplementary Figure 19. t-SNE Dimensionality Reduction Visualization of Expert Weights from*
 *NCA Battery Test Model.*

*For the NCA-material test batteries under different operating conditions in the UL dataset, we*
 *randomly selected a pre-trained model and used full-lifecycle single-cycle test samples as input.*
 *After t-SNE clustering, the expert weights output by the model were visualized, with samples from*
 *different aging stages color-mapped accordingly (see Supplementary Note 8 for details). In*
 *Supplementary Figures 1-6, as well as through SHAP and Spearman correlation coefficient analysis,*
 *the features input into the AMDP module show strong correlation with the degradation stages*
 *themselves. In the two-dimensional latent space after clustering, there is a clear boundary between*
 *healthy and aged batteries, further demonstrating the effectiveness of the MOE architecture in*
 *identifying different degradation stages and validating the rationale for using expert weights in*
 *second-life utilization decisions. Figures (a)-(d) represent three different operating conditions. “*

As shown in Fig. 4a, the degradation-router dynamically allocates weights to the expert networks
 for different battery samples. When the battery SOH is high (e.g., >95%), the model tends to assign
 higher weights to Expert 1 and Expert 2, indicating their dominant role in predicting early-stage
 degradation processes such as SEI formation and growth. When the battery SOH is low (e.g., <80%),
 the weights of Expert 4 and Expert 5 increase significantly, reflecting their specialization in
 modeling late-stage mechanisms like lithium plating and associated accelerated degradation. To
 validate the effectiveness and interpretability of expert network selection across degradation stages,
 t-SNE clustering based on expert weights (Fig. 4c) clearly shows that battery samples at different
 degradation stages form distinct clusters in the feature space. This result directly demonstrates that
 the MoE architecture can effectively capture and distinguish different degradation modes governed
 by the degree of battery aging.

**Comment 14**

The research is regarding PI-physics informed parameters, but there is no discussion of these
 parameters on the impact of aging and how they are relevant to be called physical parameters. They
 are driven by electrical tests and more statistical data.

**Response to Comment 14**

Thank you for the comment. The PI-physics informed parameters adopted in this study have clear
physical meanings and are selected based on battery electrochemical principles. In Supplementary
Figures 1-6, The authors directly display the correlation between these physical features and battery
aging. The authors have revised the paper in the following aspects:

- ● The authors have revised the title of the manuscript to avoid potential ambiguity.
- ● The authors have revised the introduction to further discuss physical modeling methods
- ● The authors have revised the discussion section to clearly outline the current limitations of the
method and the direction for future improvements:

To avoid any potential confusion for readers, the authors have revised the title of the paper.

*“PIMOE: Physically interpretable mixture of experts network for battery degradation trajectory*
*prediction amid second-life complexities.”*

The authors have revised the introduction to further discuss physical modeling methods:

*“Additionally, due to highly non-linear degradation of second-life batteries, incorporation of*
*physics knowledge into models has gained increased attention. Recent advances in sensory-based*
*measurements include X-ray imaging²⁵, electrochemical impedance, optical fiber sensing²⁶,*
*acoustic sensing²⁷, partial charging²⁸, which serves as the data-input of the data-driven methods.*
*Nevertheless, most sensing techniques remain at laboratory stage and are invasive²⁶. Meanwhile,*
*purely data-driven methods struggle to capture internal physical information of retired batteries²⁹.*
*Given the understanding of internal physical state is particularly critical for second-life*
*applications with safety-sensitive considerations, the challenge of the "black-box" nature and lack*
*of interpretability in data-driven methods for battery state prediction should be addressed. Even if*
*progresses have been made in physics-informed neural networks, the promise lies in additional and*
*explicit integration of physical laws into feature engineering process, neural network loss functions,*
*and transferability metrics. However, the complex physical constraints involve numerous*
*parameters and prior physical understanding from the modeling of retired batteries, which is still*
*hardly available post-retirement⁵. Tao et al. employed physics-informed machine learning to achieve*
*full degradation trajectory prediction using early-cycle data, reducing data requirements, while still*
*relying on historical data³⁰. In recent years, mixture-of-experts (MOE) architecture, a core*
*component in large language models, has demonstrated notable performance in learning*
*heterogeneous data representations. By decomposing complex tasks into sub-tasks handled by*

*“expert” modules with specialized task knowledge, MOE captures multi-level and nonlinear feature*
*representations. The ideal of decomposing tasks into multiple subtasks naturally aligns with coupled*
*degradation mechanisms of batteries. However, in absence of physical interpretability, the learning*
*outcomes of model are confined to purely statistical correlations and fail to reflect true physical*
*mechanisms underlying degradation processes, particularly with diverse cathode chemistries³¹,*
*historical usages³², and uncertain second-life operating conditions¹⁶. Incorporating physical*
*knowledge into the MOE architecture anchors data-driven learning outcomes to typical*
*electrochemical processes, thereby enhancing model interpretability and enabling mechanism-*
*aware degradation trajectory prediction.”*

The authors have revised the discussion section to clearly outline the current limitations of the
method and the direction for future improvements:

*“However, it must be acknowledged that PIMOE approach primarily relies on electrochemical*
*characteristics (such as voltage-capacity relationships and relaxation voltage) for macroscopic*
*diagnostics. While extensive validation confirms the physical understanding of these features^{13,33},*
*they essentially remain statistical correlations instead of physical causality. It is recommended that*
*future work should focus on non-invasive in-situ sensing signals ^{26,27,34}, such as vibration sensing,*
*strain sensing, ultrasonic signals, fiber optic sensing to enhance internal state observation for a*
*physics-informed modeling, instead of physically interpretable presented in this work. Building a*
*“physics-statistics” fusion framework has the potential to improve the interpretability of PIMOE ²⁹,*
*for example, parameters from battery electrochemical or equivalent circuit models could be updated*
*online as an intermediate physical interpretation layer for degradation trajectory prediction.*
*Integrating primary governing equations of battery modeling with data-driven methods would also*
*provide physical interpretability³⁵. ”*

**Comment 15**

What is the justification for Figure 4d, where the SEI and Li-plating are plotted separately, while
these events could happen in the same cell, depending on the activated mechanisms.

**Response to Comment 15**

Thank you for your comment. As you rightly pointed out, SEI growth and lithium plating may occur
sequentially within the same battery through different activation mechanisms. The authors
acknowledge that categorizing battery conditions based solely on dominant degradation

mechanisms is oversimplified and overly broad. The original separation of the three mechanisms in
 Figure 4d did causes ambiguity. The authors have revised Figure 4d to eliminate this
 misunderstanding and have further clarified this point in the main text.

“ In Fig 4d, for the 130 UL dataset batteries retired at different SOH levels (classified solely based
 on expert weights, see Supplementary Note 7 for criteria), those retired at 95% SOH achieved an
 "excellent" classification confidence of 91.5%, while batteries retired at 75% SOH received a
 "scrap" rating of 96%. This further demonstrates that the model utilizes different primary experts
 for prediction based on the dominant degradation mechanisms at various stages, thereby enabling
 the reverse classification of retired batteries at different degradation stages according to expert
 selection. ”

**Response to Reviewer #2**

This is an interesting paper addressing the challenge of predicting battery degradation trajectories
in second-life applications using a novel physics-informed mixture of experts (PIMOE) architecture.
The manuscript is generally well-written, and the proposed approach is innovative, combining
domain knowledge with machine learning for interpretable predictions. The results are compelling
and relevant to both academia and industry.
However, while the paper shows promise, there are several important issues that must be addressed
before it can be considered for publication.

Dear Respected Reviewer,

Dear Reviewer,

The authors respond to your suggestions point by point and carefully revise the manuscript and
supplementary information, where you can find corresponding revisions using “*track changes*”
mode. Changes are colored in *blue* in revised clean version. The authors truly hope that the responses
appropriately address your justified concerns, and that revised manuscript meets your expectations.

**Comment 1**

The approach assumes that future load conditions are known in advance, which is a strong
assumption in real-world second-life scenarios. Although this is briefly acknowledged in the
discussion, a more thorough examination of the limitations and implications of this assumption is
warranted. Furthermore, scenarios involving uncertain or stochastic future use should be considered
or at least simulated.

**Response to Comment 1**

Thank you for the comment. You pointed out the limitation of the assumption that “future load
conditions are fully known” in real-world second-life battery applications. However, it is important
to note that the degradation trajectory of a battery is inherently path-dependent, as its future
degradation rate and maximum usable capacity are closely influenced by operating conditions.
Although the proposed method can be extended to long-term degradation trajectory prediction and
has been extensively validated across different prediction horizons, applying short-term predictions
in deployment scenarios can greatly mitigate the impact of uncertainty or randomness in future
operating conditions on prediction accuracy. The authors have comprehensively analyzed
representative application scenarios of retired batteries to verify the effectiveness of the proposed

method and further supplemented evaluations on deeply degraded retired battery datasets to
minimize the influence of random variability.

The authors have revised the paper in the following aspects:

- ● The authors further analyzed a newly released retired battery dataset to more comprehensively
evaluate the superiority of the proposed method under different second-life scenarios.
- ● The authors have added descriptions of the newly introduced dataset and revised Figure 2 to
provide more comprehensive coverage of retired battery scenarios.
- ● The authors have included performance analyses of the new dataset and modified Figure 3 to
evaluate the prediction performance of deeply degraded batteries
- ● The authors have added performance evaluations across different SOH intervals in the
uncertainty analysis section.
- ● The authors have added detailed information of the new dataset in Supplementary Table 3.
- ● The authors have added a detailed analysis of the limitations of this assumption, along with
potential improvements and extensions for future methodologies

*“ The LSD dataset contains 86 commercial batteries with $\text{LiNi}_{0.5}\text{Co}_{0.2}\text{Mn}_{0.3}\text{O}_2$ cathodes and*
*graphite anodes, These 2.4 Ah batteries (3.7 V nominal) underwent two-phase testing. In Phase 1,*
*the batteries were grouped into 16 distinct charge–discharge protocols to induce diverse*
*degradation behaviors as the SOH decreased from 100% to 80% (details in Supplementary Table*
*3). In Phase 2, all 86 cells were re-cycled from 80% to 50% SOH under a unified low-rate protocol*
*(0.5 C charge / 0.2 C discharge) to simulate stationary energy-storage operation. Degradation*
*trajectories in Fig. 2b demonstrate Phase 2 cycling variability, We predict the deep degradation*
*trajectories of second-phase batteries under unknown and varying historical operating conditions. ”*

**Fig. 2 Dataset Description.** (a) Summary of the UL dataset testing conditions and the degradation
 curves of the batteries. (b) Summary of the LSD dataset testing conditions and the degradation
 curves of the batteries. (c) Summary of the TPSL dataset testing conditions and the degradation
 curves of the batteries. (d) Capacities distribution of the cells at different use conditions. (e) The
 charging curves with random initial state of charge and the 30-minute relaxation voltage curves.

The authors have added the following description in the Model performance and generalization
 capability:

“Fig. 3b reveals that in TPSL data, baseline methods displayed limited capability in capturing
 long-term trends after future load changes and completely failed during initial load transitions. In
 contrast, PIMOE effectively identified transitional characteristics between phases through future
 usage condition integration. Supplementary Figs.11-14 further demonstrate PIMOE's successful
 learning of relationships between future operating conditions and capacity performance. In the LSD
 dataset, the baseline models exhibited larger errors in predicting the later stages of degradation

trajectories. This is mainly due to the unknown and varying operating conditions in the first phase,
which resulted in highly heterogeneous degradation behaviors during deep aging. In contrast, our
proposed method, benefiting from the AMDP module, achieved more stable and reliable prediction
performance.

[revised manuscript text omitted]

The authors have added the following description in the Supplementary Table 3:

**“Supplementary Table 2. Technical Specifications of LSD Dataset.**

The table presents the materials, operating conditions, nominal capacities, cutoff voltages, and
 number of batteries used in our experimental dataset. During phase I, as the SOH of the batteries
 declined from 100% to 80%, 86 cells were divided into 16 groups, with each group subjected to a
 distinct charge–discharge protocol. In phase II, as the SOH further decreased from 80% to 50%, all
 86 cells were cycled under a unified protocol. The dataset is designated as the LSD Dataset.

*In the phase 2 dataset, although all batteries were operated under identical cycling conditions,*
*differences in the loading conditions during their first-life usage led to varying impacts on internal*
*degradation mechanisms, resulting in significantly divergent degradation trajectories during the*
*second-life phase. This phenomenon highlights the profound influence of historical usage conditions*
*on the long-term health evolution of batteries and provides an ideal validation scenario for*
*evaluating the model's predictive capability under unknown historical conditions and deep*
*degradation states. We utilized only the second-phase dataset to simulate the model's ability to*
*predict degradation trajectories of deeply aged, retired batteries in deployment scenarios where*
*historical data are unavailable.*

Cell types	Material	Working conditions			Nominal capacity (Ah)	Cut-off voltage (V)	Number of cells
		Charging C rate	Discharging C rate	Temperature (°C)			
Phase 1	LiCoO ₂ and LiNi _{0.5} Co _{0.2} Mn _{0.3} O ₂ / graphite anodes	0.5	1	25	2.4	3.0-4.2	8
		0.5	1	25			8
		0.5	1	25			6
		1	1	25			8
		1.5	1	25			8
		2	1	25			4
		0.5	2	25			4
		0.5	3	25			4
		1	1	25			4
		1	1	25			4
		1.5	1	25			4
		1.5	1	25			4
		2	3	25			4
		0.5	0.2	25			3
0.5	0.5	25	7				
0.5	2	25	4				
Phase 2		0.5	1	25			86

In the discussion section, the authors have added a detailed analysis of the limitations of this
assumption, along with potential improvements and extensions for future methodologies.

*“Even though the assumption that future use conditions are available is reasonable for verifying*
*the effectiveness of PIMO, and projects such as home energy storage and integrated photovoltaic-*
*storage systems often have pre-defined operational plans, the actual operating conditions of battery*

*energy storage systems remain highly dependent on application scenarios and are difficult to predict.*
*Although the paper has discussed the importance of introducing future use conditions in the FORNN*
*structure, the analysis of the uncertainty of future use conditions needs to be extended to highly*
*random and time-varying use scenarios, Future work could focus on introducing load conditions at*
*finer temporal resolutions to ensure the validity of predicted load impacts on batteries. more*
*importantly, with statistical confidence margins, of the battery energy storage system to ensure the*
*safe and duration operation.”*

**Comment 2**

While the model is claimed to be "physics-informed," its interpretability largely relies on
correlations extracted from capacity-voltage and relaxation curves. These features, while derived
from empirical electrochemical knowledge, do not truly integrate physics-based modeling such as
differential equations or parameter estimation from equivalent circuit models. A clearer distinction
between physics-informed and statistically correlated is needed.

**Response to Comment 2**

Thank you for the comment. Expression of "physical informativeness" in the initial draft could
indeed lead to misunderstandings. As you pointed out, although the interpretability of model relies
on features extracted from the capacity-voltage and relaxation curves, these features are based on
electrochemical empirical knowledge, but they are not directly embedded in physical equations
describing the battery's dynamic processes or the parameter estimation of equivalent circuit models.
The current model's physics-informed " is more about extracting correlation features from empirical
data rather than fully integrating traditional physical modeling methods. This distinction between
"physical physics-informed " and "statistical correlation" should indeed be further clarified in the
paper. The authors have revised the paper in the following aspects:

- ● The authors have revised the title of the manuscript to avoid potential ambiguity.
- ● The authors have revised the introduction to further discuss physical modeling methods
- ● The authors further illustrate the correlation between degradation mechanisms at different
stages of degradation and the expert network by analyzing the IC curves of the dataset.
- ● The authors further validated the rationality of these physical features through SHAP analysis
and Spearman correlation coefficients.
- ● The authors have revised the discussion section to clearly outline the current limitations of the
method and the direction for future improvements:

The authors have revised the title of the paper:

*“PIMOE: Physically interpretable mixture of experts network for battery degradation trajectory*
*prediction amid second-life complexities. ”*

The authors have revised the introduction to further discuss physical modeling methods:

[revised manuscript text omitted]

The authors further illustrate the correlation between degradation mechanisms at different stages of
degradation and the expert network by analyzing the IC curves of the dataset.

*“ Supplementary Figure 28. IC Aging Curve Analysis of the Dataset*

*This section aims to demonstrate the rationality and universality of dividing battery aging into*
*multiple stages using IC curves, and to explain how these parameters reveal the battery degradation*
*mechanisms and reflect their physical significance. This assumption is based on numerous studies*
*on lithium battery degradation patterns. For instance, existing research suggests that during the*
*battery's lifecycle, the formation and thickening of the SEI layer affect the capacity degradation in*
*the early stages, while lithium-ion deposition typically occurs in the later stages of battery use and*
*is one of the main causes of the sharp capacity decline.*

*IC curves reveal internal changes in the battery by showing the subtle variations between battery*
*capacity and voltage, especially during the aging process, where the SEI layer growth, lithium-ion*
*loss, and internal electrochemical reactions significantly impact the IC curve. As shown in the figure,*
*different colors represent different SOH samples. In the early charging and discharging cycles, the*
*curve displays sharp peaks, indicating the battery is in a healthy state with significant and rapid*
*capacity changes, mainly reflecting the formation of the SEI layer and surface chemical reactions.*
*As the cycle count increases, the peaks of the IC curve become more rounded and shift to the right,*
*and longer flat sections appear, indicating that the battery's reversible capacity is gradually*
*decreasing. When the battery enters a more severe degradation phase, the peaks of the IC curve*
*gradually decrease. At this point, degradation mechanisms such as lithium-ion deposition and loss*
*of active materials dominate the capacity decline. The widening of the peaks in the curve and the*
*sharp capacity drop indicate that the battery's energy storage capacity is severely compromised.*

In Supplementary Figures 1-6, we directly display the correlation between these physical features
 and battery aging. Furthermore, we have further validated the rationality of these physical features
 through SHAP analysis and Spearman correlation coefficients.

The authors have made the following modifications to the Supplementary Figure 22:

*“ To quantify the influence of input features on the model's predictions and elucidate the*
 *underlying physical mechanisms, we employed SHapley Additive exPlanations (SHAP) analysis and*
 *Spearman's rank correlation analysis. As shown in Fig. a, the SHAP analysis based on NCA+NCM*
 *battery data reveals the differential contributions of the 12 physics-informed features to the final*
 *prediction outcome. Concurrently, Spearman's correlation analysis in Fig. b demonstrates that most*
 *features exhibit statistically significant correlations with SOH. These two analytical results*
 *corroborate each other, collectively confirming the robustness and reliability of the selected physical*
 *features in effectively characterizing the battery degradation state.*

Based on the strong correlation between the selected input features of the AMDP module and SOH,
 the degradation gating network can effectively distinguish data samples from different degradation
 stages to make corresponding predictions. By directly performing t-SNE clustering on the weights
 of different experts, PIMOE shows clear clustering boundaries for samples from different
 degradation stages, further demonstrating the effectiveness of the MoE architecture in identifying
 different degradation stages.

“ *Supplementary Figure 19. t-SNE Dimensionality Reduction Visualization of Expert Weights from*
 *NCA Battery Test Model.*

*For the NCA-material test batteries under different operating conditions in the UL dataset, we*
 *randomly selected a pre-trained model and used full-lifecycle single-cycle test samples as input.*
 *After t-SNE clustering, the expert weights output by the model were visualized, with samples from*
 *different aging stages color-mapped accordingly (see Supplementary Note 8 for details). In*
 *Supplementary Figures 1-6, as well as through SHAP and Spearman correlation coefficient analysis,*
 *the features input into the AMDP module show strong correlation with the degradation stages*
 *themselves. In the two-dimensional latent space after clustering, there is a clear boundary between*
 *healthy and aged batteries, further demonstrating the effectiveness of the MOE architecture in*
 *identifying different degradation stages and validating the rationale for using expert weights in*
 *second-life utilization decisions. Figures (a)-(d) represent three different operating conditions. “*

The authors have revised the discussion section to clearly outline the current limitations of the
 method and the direction for future improvements:

*“However, it must be acknowledged that PIMOE approach primarily relies on electrochemical*
 *characteristics (such as voltage-capacity relationships and relaxation voltage) for macroscopic*
 *diagnostics. While extensive validation confirms the physical understanding of these features^{13,33},*
 *they essentially remain statistical correlations instead of physical causality. It is recommended that*
 *future work should focus on non-invasive in-situ sensing signals^{26,27,34}, such as vibration sensing,*
 *strain sensing, ultrasonic signals, fiber optic sensing to enhance internal state observation for a*
 *physics-informed modeling, instead of physically interpretable presented in this work. Building a*
 *“physics-statistics” fusion framework has the potential to improve the interpretability of PIMOE²⁹,*
 *for example, parameters from battery electrochemical or equivalent circuit models could be updated*
 *online as an intermediate physical interpretation layer for degradation trajectory prediction.*
 *Integrating primary governing equations of battery modeling with data-driven methods would also*
 *provide physical interpretability³⁵. ”*

**Comment 3**

The paper compares its approach to two baselines (Informer and PatchTST), which are relevant
 time-series models. However, additional baselines from battery-specific modeling domains—such

as hybrid physics-informed neural networks or direct state-of-health estimation models—would
strengthen the empirical evidence. Also, it is unclear whether the baselines were retrained under the
same conditions, which affects reproducibility.

**Response to Comment 3**

Thank you for the comment. It should be noted that most existing research focuses on SOH in in-
vehicle BMS scenarios, where degradation trajectory prediction tasks require a substantial amount
of historical data as input. These models are ineffective for retired battery scenarios. The authors
have revised the paper in the following aspects:

- ● The authors provide detailed information on the training settings and experimental conditions
of these baseline models in the revised version.
- ● The authors have provided the original code and training code for all models.

To ensure the reproducibility of the research results, the authors will provide detailed information
on the training settings and experimental conditions of these baseline models in the revised version,
ensuring that all models are compared under unified training conditions. The authors have provided
the original code and training code for all models. The authors will also provide more information
on the baseline model training process so that other researchers can more easily replicate these
results.

*“Supplementary Note 4. Implementation details of baselines and PIMOE.*

PatchTST³⁷ is a Transformer-based model utilizing patching technique. It enables effective pre-
training and transfer learning across datasets. The source code is available at (<https://github.com/yuqinie98/PatchTST>).

Informer³⁸ is a Transformer-based model specifically designed for long-sequence time-series
forecasting. It introduces a ProbSparse self-attention mechanism to improve computational
efficiency while maintaining prediction accuracy. The model also features a memory-efficient
architecture that enables effective handling of large-scale datasets. The official implementation is
available at (<https://github.com/zhouhaoyi/Informer2020>).

Both PatchTST and Informer were implemented according to their official source codes. For the
Informer model, due to the unavailability of historical data for retired batteries, temporal
information is reflected in the charging curve of the current cycle. We retained the encoder part and
used an MLP as the decoder to serve as a predictor for the degradation trajectory. For the PatchTST

model, the continuous sampling points of the charging curve were similarly treated as patch
information. Where applicable, the baseline models and our proposed method used the same
hyperparameters. We tuned the number of layers with reference to ^{14,39}, the neuron number in hidden
linear layers from ⁴⁰, the weight decay in Adam optimizer within [0, 0.0001], and the number of
attention heads for multi-head attention from ^{41,42}. The patch size was set to 3 based on the Q-V
curve length³⁷. The batch size was set to [8, 16, 64, 128]. We applied for a learning rate of 0.001
across all datasets and selected the best-performing model on the validation set as our final model.”

”

**Comment 4**

The following two recent and highly relevant works in the field of imitation learning-based
degradation modeling are not cited and must be referenced:

• Pozzi, A., Incremona, A. & Toti, D. "Imitation learning-driven approximation of stochastic control
models". Applied Intelligence, 55, 838 (2025). <https://doi.org/10.1007/s10489-025-06704-x>

• Pozzi, A., Incremona, A., Toti, D. "Neural Network-Based Imitation Learning for Approximating
Stochastic Battery Management Systems," IEEE Access, vol. 13, pp. 71041-71052, 2025.
<https://doi.org/10.1109/ACCESS.2025.3563300>

These works are directly relevant to the idea of modeling stochastic system dynamics using data-
driven expert systems and should be included for context and completeness.

**Response to Comment 4**

Thank you for the comment. These two articles are highly relevant to our research, and the authors
have already cited them in the revised version.

*“With acknowledging the assumption that future use conditions can be assumed, it is reasonable for*
*verifying the effectiveness of PIMOE under predefined uncertain scenarios with considerable*
*uncertainties, specifically, consistent operating histories, deeply aged batteries with unknown prior*
*use and varying second-life conditions with 95 use conditions being validated. However, despite*
*practical applications such as home energy storage and integrated photovoltaic-storage systems*
*often have pre-defined operational plans, the actual operating conditions of battery energy storage*
*systems remain even more uncertain and are difficult to predict ^{36,43}. Although this work has*
*discussed the importance of introducing future use conditions in the FORNN structure, the analysis*
*of the uncertainty of future use conditions needs to be extended to highly random and time-varying*
*use scenarios. Thus, future work could focus on introducing load conditions at fine temporal*
*resolutions to ensure the validity of predicted use conditions of retired batteries. Another aspect is*

*that random initial SOC is an important factor in the data availability of retired batteries. Although*
*the effectiveness of PIMOE in predicting degradation trajectory under different initial SOC*
*conditions has been validated, the initial SOC of retired batteries is more uncontrollable due to the*
*time-consuming nature of acquisition such information. Future work suggests considering SOC*
*normalization during the feature extraction phase or introducing an SOC-aware routing mechanism*
*in the model, to better handle battery degradation difference under different SOC states⁴⁴. Although*
*the paper has considered four very different application conditions, namely UL, LSD, TPSL-*
*Random, TPSL-fixed, given that the second-life application of the battery is safety-critical, more*
*extreme conditions need to be investigated. This is because the robustness of the dataset to non-*
*steady-state use conditions, such as the battery with thermal runaway or internal short circuits, has*
*not been included. Considering the bias of the training data is crucial, and it is necessary to assess*
*the uncertainty and confidence intervals of the predicted degradation trajectories, as UL and TPSL*
*data hardly cover all environmental fluctuations under actual service conditions, such as thermal*
*non-uniformity at the battery pack level. Future work is suggested to collect real-world variables*
*that include different manufacturers, degradation mechanisms, and cell-to-cell variations.”*

**Comment 5**

The current model provides deterministic predictions. In safety-critical applications like battery
reuse, it is essential to assess the uncertainty or confidence bounds of the predicted degradation
trajectories. The paper mentions this briefly in the discussion but does not quantify it in the results.
A demonstration or discussion of uncertainty-aware predictions (e.g., via ensembles or Bayesian
methods) is encouraged.

**Response to Comment 5**

Thank you for the comment. The authors fully agree on the importance of assessing prediction
uncertainty in safety-critical applications such as battery second-life utilization, as well as
considering the inherent uncertainty in deep learning model outputs. To ensure result stability, the
authors conducted 10 repeated experiments for all trials using different random seeds. Furthermore,
the authors performed a detailed analysis of how various uncertainty factors—including model
hyperparameters, training data volume, and prediction length—affect performance. That said, the
authors recognize the necessity of quantitatively characterizing uncertainty in the results. Therefore,

the authors have now comprehensively quantified the uncertainty for all experimental outcomes.

The authors have revised the paper in the following aspects:

● The authors have made the following modifications to the discussion:

● The authors have performed a detailed quantitative analysis and made the following
modifications to the Supplementary Table 4:

The authors have made the following modifications to the discussion:

“Although the paper has considered four very different application conditions, namely UL, LSD,
TPSL-Random, TPSL-fixed, given that the second-life application of the battery is safety-critical,
more extreme conditions need to be investigated. This is because the robustness of the dataset to
non-steady-state use conditions, such as the battery with thermal runaway or internal short circuits,
has not been included. Considering the bias of the training data is crucial, and it is necessary to
assess the uncertainty and confidence intervals of the predicted degradation trajectories, as UL and
TPSL data hardly cover all environmental fluctuations under actual service conditions, such as
thermal non-uniformity at the battery pack level. Future work is suggested to collect real-world
variables that include different manufacturers, degradation mechanisms, and cell-to-cell variations.”

Additionally, the authors have performed a detailed quantitative analysis and made the following
modifications to the Supplementary Table 5:

*“Supplementary Table 5. The performance of the model in Four different usage scenarios.*

*Here, we demonstrate the predictive performance of the PIMOE model under three distinct*
*scenarios: UL representing the full lifecycle under constant operating conditions, LSD representing*
*the deep degradation dataset, and TPSL corresponding to repurposed usage with varying operating*
*conditions (specific naming conventions are detailed in Supplementary Table 1). To ensure*
*experimental rigor, all results represent averages from 10 randomized trials. Notably, the models*
*exclusively utilized partial cycle data collected in situ as input to predict 50-cycle capacity*
*degradation trajectories under uncertain future operating conditions.*

Error Metric	MAPE (%)	RMSE (%)	MAE (%)
UL	0.520852	1.878831	1.404282

TPSL-Arbitrary	2.957535	4.277484	3.326660
TPSL-Fixed	2.806381	5.522538	3.857384
LSD	1.809982	3.655229	2.805560

**Response to Reviewer #3**

This a well written paper on a general topic of interest to the battery community. I can recommend
 publication if the authors address the following comments.

Dear Reviewer,

The authors respond to your suggestions point by point and carefully revise the manuscript and
 supplementary information, where you can find corresponding revisions using “*track changes*”
 mode. Changes are colored in *blue* in revised clean version. The authors truly hope that the responses
 appropriately address your justified concerns, and that revised manuscript meets your expectations.

**Comment 1**

Spearman coefficient & SHAP values. The authors rely on their physics-informed router + experts
 structure, but it would strengthen the paper to validate feature importance explicitly:

- • A simple Spearman correlation analysis between input features and outputs (SOH, degradation
 modes) could confirm monotonic relationships and highlight redundancies among the 12 features.
- • A SHAP (Shapley Additive Explanations) analysis on the trained neural model could provide
 insight into which features the network relies on most. This would help connect the “physics-
 informed” explanation with the actual feature usage inside the NN.
- • Suggestion: Add a supplementary analysis of feature importance (Spearman + SHAP) to improve
 interpretability and confidence in the model.

**Response to Comment 1**

Thank you for the comment. This suggestion has significantly enhanced the rigor and interpretability
 of our research. Following your advice. The authors have revised the paper in the following aspects:

- ● The authors have conducted a comprehensive supplementary analysis involving both
 Spearman's coefficient and SHAP values.
- ● The authors have made the following modifications to the Supplementary Figure 22:

The authors have conducted a comprehensive supplementary analysis involving both Spearman's
coefficient and SHAP values.

“ *Supplementary Note 12. SHAP Interpretability Analysis*

*This section aims to elucidate the contribution of physics-informed features to the model outcomes*
*through SHAP analysis. SHAP is an interpretability method grounded in cooperative game theory,*
*designed to explain the predictions of machine learning models. Its core concept involves*
*calculating the marginal contribution of each feature to individual prediction outcomes, thereby*
*equitably allocating feature importance. SHAP values provide local interpretability, clearly*
*demonstrating how each feature drives the model's prediction from the baseline value to the final*
*output. This study employs SHAP analysis to reveal the decision-making basis of the constructed*
*model and to identify the key factors and their direction of influence that drive the expert gating*
*outcomes at different stages.*

*The SHAP value Φ_i quantifies the contribution of feature i to the output $v(N)$, derived from its*
*marginal contribution.*

$$\Phi_i = \sum_{S \in N/\{i\}} \frac{|S|! (n - |S| - 1)!}{n!} [v(S \cup \{i\}) - v(S)]$$

*where S is a subset of the full feature set N , and $|S|$ indicates the size (number of features) of the*
*subset. The function is the payoff function for the subset S .*

*The sum of the SHAP values of all features equals the difference between the predicted value for the*
*sample and the baseline value (the expected prediction value of the model)*

$$g(z') = \Phi_0 + \sum_{i=1}^M \Phi_i z'_i,$$

*Here, Φ_0 represents the baseline prediction of the model, Φ_i denotes the SHAP value of feature ,*
*and M indicates the total number of features.*

The authors have made the following modifications to the Supplementary Figure 22:

“ *To quantify the influence of input features on the model's predictions and elucidate the*
*underlying physical mechanisms, we employed SHapley Additive exPlanations (SHAP) analysis and*
*Spearman's rank correlation analysis. As shown in Fig. a, the SHAP analysis based on NCA+NCM*
*battery data reveals the differential contributions of the 12 physics-informed features to the final*
*prediction outcome. Concurrently, Spearman's correlation analysis in Fig. b demonstrates that most*

*features exhibit statistically significant correlations with SOH. These two analytical results*
 *corroborate each other, collectively confirming the robustness and reliability of the selected physical*
 *features in effectively characterizing the battery degradation state.*

Comment 2

Reduction in Number of features. The manuscript states that the input consists of 12 features derived
 from partial charge + relaxation voltage profiles. While 12 is not excessive for modern ML, some
 may be correlated or redundant.

• Suggestion: Discuss whether feature selection or dimensionality reduction (e.g., PCA, correlation
 filtering, or SHAP pruning) could simplify the model without significant loss in accuracy. This also
 makes the physics-informed routing more interpretable.

**Response to Comment 2**

Thank you for the comment. This suggestion is of significant value for improving the efficiency of
 model. In supplementary Figure 25, The authors removed the features extracted from the QV curves
 and relaxation curves separately to illustrate the impact of feature quantity on the results. As you
 mentioned, conducting analyses of feature importance and correlation is indeed necessary. The
 authors have revised the paper in the following aspects:

- ● The authors have modified Supplementary Figure 25 to illustrate the relationship between
 feature quantity and model accuracy.
- ● The authors have added Supplementary Figure 27 to evaluate the impact on model accuracy
 after reducing features with low correlation.

The authors have made the following modifications to Supplementary Figure 25:

*“Supplementary Figure 25. Impact of Different Physics-informed Feature on Model Performance.*

Here we present the performance changes of the model when reducing physics-informed features.
"Both" denotes the complete model incorporating both capacity-voltage curves and relaxation-
voltage curves. "no QV features" indicates the removal of six features extracted from capacity-
voltage curves, while "no RV features" represents the exclusion of six features derived from
relaxation-voltage curves. The results show that removing RV features leads to relatively greater
performance degradation, suggesting that relaxation-voltage features may more effectively guide
the degradation routing. Nevertheless, the model maintains relatively stable performance overall,
demonstrating that the proposed PIMOE does not depend on any single specific feature to achieve
good performance, though multi-dimensional physics-informed features generally yield better
results. "

Based on Spearman correlation analysis and SHAP value analysis, the authors conducted further
feature analysis and selection. Specifically, the authors removed 6 features with low correlation in
both analyses to assess the impact of feature reduction on model performance.

The authors have added the following in supplementary Figure 28:

*" Supplementary Figure 28. Performance Analysis After Feature Reduction*

*The input of this paper consists of 12 features extracted from partial charging and relaxation voltage*
*curves. Although 12 features may not seem many for modern machine learning methods, there may*
*be issues with high correlation or redundancy among them. Based on SHAP and Spearman*
*correlation analysis, we conducted further feature analysis and selection. Specifically, we removed*
*6 features that showed low correlation in both analyses to assess the impact of feature reduction on*
*model performance. Each experiment was repeated 10 times to ensure rigor, with different colors*
*representing the performance of the original model and the model with reduced feature count. The*
*experimental results show that after removing the six low-correlation features, the model*

*performance experienced a slight increase. This indicates that features with higher correlations*
*have a more significant impact on the model's performance within the selected dataset. In*
*deployment scenarios with limited computational resources, feature reduction can help strike an*
*effective balance between performance and computational efficiency. “*

**Comment 3**

Effect of random SOC starting point. The work does not discuss how the initial SOC (random
starting point of partial charging) influences the extracted features. In practice, this can strongly
affect the relaxation trajectory and hence the prediction accuracy.

• Ignoring this factor may lead to optimistic performance compared to real-world deployment, where
SOC is not always controlled.

• Suggestion: Even if beyond the current study's scope, add a short discussion of this limitation and
how future work might address it (e.g., SOC normalization, SOC-aware routers). This would show
awareness of practical challenges.

**Response to Comment 3**

Thank you for the comment. Random initial SOC has a significant impact on feature extraction and
prediction accuracy, especially when the initial SOC is uncontrollable in real-world deployments.

Therefore, in the main text, The authors have conducted targeted analytical experiments to assess
the prediction performance of the proposed approach under random initial SOC conditions.

However, as you mentioned, this analysis is still somewhat idealized. The SOC normalization and
SOC-aware routing mechanism you referred to would effectively address these issues. The authors
have revised the paper in the following aspects:

● The authors have revised the introduction to more specifically describe the challenges posed
by random SOC conditions.

● The authors have revised the discussion section to clearly outline the current limitations of the
method and the direction for future improvements:

The authors have revised the introduction to more specifically describe the challenges posed by

random SOC conditions.

*“Degradation trajectory is a widely adopted indicator for battery degradation characterization,*
*which can be critical to the safety performance during extended reusing or repurposing process, i.e.,*
*second-life of the retired batteries¹⁻⁴. For example, retired batteries undergo degradation trajectory*
*evaluation before second-life use based on the cell, module or pack-level application requirements^{5,6}.*
*Conventional non-data-driven methods relying on destructive disassembly or full-cycle capacity*
*tests^{7,8}, inflicting extra damage or labor investments. Notably, the economic feasibility of these state*
*measurement methods of retired batteries are impractical due to prohibitive time, momentary, and*
*environmental costs at scale⁹⁻¹². Data-driven methods have demonstrate promises in predicting*
*degradation trajectories using non-destructive electrical signal data, but they typically require 5%*
*-40% of full lifecycle historical data¹³⁻¹⁵. Moreover, uncertain second-life use conditions can*
*significantly differ from those in first-life, rendering conventional battery degradation trajectory*
*prediction models ineffective, given that the data used for model training were built on constant-*
*condition tests or typical dynamical cycling tests while the degradation trajectory is highly path-*
*dependent¹⁶⁻¹⁸. Lu et al. leveraged at least one full cycle of capacity-voltage curves to predict*
*degradation trajectories under uncertain future operating scenarios by learning the relationship*
*between use condition and battery capacity¹⁶. Yet, the method encounters limitations due to the*
*stochasticity in state of charge (SOC) of retired batteries and the demanding requirement for full*
*discharge-charge data. Moreover, the unavailability of historical cycling data of retired batteries*
*complicates the tracing of initial conditions of degradations, such as impedance rise, loss of lithium*
*inventory, and loss of active material¹⁹⁻²², challenging degradation trajectory prediction*
*effectiveness. At the policy level, global initiatives such as the Battery Data Genome Initiative²³ and*
*the European Commission’s July 4, 2025 Delegated Regulation on battery recycling²⁴ aim to*
*enhance data standardization, traceability, and material recovery across the battery lifecycle.*
*However, the policy advances still face practical challenges due to frequent ownership change and*
*uncertain second-life use conditions retired batteries, emphasizing the need for degradation*
*prediction frameworks that rely solely on field-accessible data and adapt to diverse operational*
*environments.*

*Additionally, due to highly non-linear degradation of second-life batteries, incorporation of physics*

*knowledge into models has gained increased attention. Recent advances in sensory-based*
*measurements include X-ray imaging²⁵, electrochemical impedance, optical fiber sensing²⁶,*
*acoustic sensing²⁷, partial charging²⁸, which serves as the data-input of the data-driven methods.*
*Nevertheless, most sensing techniques remain at laboratory stage and are invasive²⁶. Meanwhile,*
*purely data-driven methods struggle to capture internal physical information of retired batteries²⁹.*
*Given the understanding of internal physical state is particularly critical for second-life*
*applications with safety-sensitive considerations, the challenge of the "black-box" nature and lack*
*of interpretability in data-driven methods for battery state prediction should be addressed. Even if*
*progresses have been made in physics-informed neural networks, the promise lies in additional and*
*explicit integration of physical laws into feature engineering process, neural network loss functions,*
*and transferability metrics. However, the complex physical constraints involve numerous*
*parameters and prior physical understanding from the modeling of retired batteries, which is still*
*hardly available post-retirement⁵. Tao et al. employed physics-informed machine learning to achieve*
*full degradation trajectory prediction using early-cycle data, reducing data requirements, while still*
*relying on historical data³⁰. In recent years, mixture-of-experts (MOE) architecture, a core*
*component in large language models, has demonstrated notable performance in learning*
*heterogeneous data representations. By decomposing complex tasks into sub-tasks handled by*
*"expert" modules with specialized task knowledge, MOE captures multi-level and nonlinear feature*
*representations. The ideal of decomposing tasks into multiple subtasks naturally aligns with coupled*
*degradation mechanisms of batteries. However, in absence of physical interpretability, the learning*
*outcomes of model are confined to purely statistical correlations and fail to reflect true physical*
*mechanisms underlying degradation processes, particularly with diverse cathode chemistries³¹,*
*historical usages³², and uncertain second-life operating conditions¹⁶. Incorporating physical*
*knowledge into the MOE architecture anchors data-driven learning outcomes to typical*
*electrochemical processes, thereby enhancing model interpretability and enabling mechanism-*
*aware degradation trajectory prediction."*

**We further discuss potential future solutions in the discussion section.**

*" Another aspect is that random initial SOC is an important factor in the data availability of*
*retired batteries. Although the effectiveness of PIMOE in predicting degradation trajectory under*
*different initial SOC conditions has been validated, the initial SOC of retired batteries is more*

*uncontrollable due to the time-consuming nature of acquisition such information. Future work*
*suggests considering SOC normalization during the feature extraction phase, or introducing an*
*SOC-aware routing mechanism in the model, to better handle battery degradation difference under*
*different SOC states. ”*

**Comment 4**

Aging trajectory. One of the key assumptions in the developed model is that the aging trajectory can
be divided into three distinct zones: SEI formation, SEI thickening, and lithium plating. However,
this assumption lacks sufficient rationale or supporting evidence for the aging datasets analyzed.
The authors should elaborate on this degradation classification, ideally providing experimental or
model-based justification. Furthermore, it would be helpful if the authors discuss the sensitivity of
their methodology/result to the choice of degradation mechanism. For instance, how would the
approach change if loss of active material were considered as one of the dominant degradation
pathways?

**Response to Comment 4**

Thank you for the comment. First, regarding the degradation mechanism classification of SEI
formation, SEI thickening, and lithium-ion deposition, this hypothesis is based on extensive research
on lithium battery degradation patterns. For example, studies (such as Tao et al. 2023, Pinson et al.
2013) have proposed that during the battery's lifecycle, the formation and thickening of the SEI
layer impact the battery's capacity degradation in the early stages, while lithium-ion deposition
typically occurs in the later stages of the battery's use and is one of the main causes of sharp capacity
decline. However, the authors recognize the necessity of specific experimental data or direct
validation for this dataset. Therefore, the authors have revised the paper in the following aspects:

- ● The authors add references to relevant literature and discuss how experimental data could
further validate the classification of these degradation mechanisms.
- ● The authors conducted an incremental capacity curve analysis on the utilized datasets to
validate the universality of the proposed degradation mechanisms.
- ● The authors have revised the discussion section to clearly outline the current limitations of the
method and the direction for future improvements:

The authors have made the following modifications to the main text:

*“Existing research demonstrates that battery degradation can be divided into three typical while*
*distinct degradation phases: SEI formation, SEI thickening, and lithium plating^{1,45}. During the*
*battery's lifecycle, the formation and thickening of the SEI layer impact the battery's capacity*
*degradation in early stages, while lithium-ion deposition typically occurs in the later stages of the*
*battery's use and is one of the main causes of sharp capacity decline. However, it is important to*
*note that the capacity knee at the end of life (EOL) is not always directly related to lithium plating.*
*In some cases, the capacity decline at the EOL may also be influenced by other degradation*
*mechanisms, such as SEI thickening, loss of active material, or electrolyte degradation, which can*
*also contribute to the sharp capacity drop observed in later stages. The degradation patterns across*
*these phases exhibit significant differences in both available capacity and subsequent aging rates,*
*especially in the slope of the degradation trajectory, with each phase reflecting distinct degradation*
*behaviors.^{46,47} **Supplementary Figure 28** illustrates that the definition of degradation stages and*
*mechanisms can be regarded as typical and reasonable. ”*

The authors further illustrate the correlation between degradation mechanisms at different stages of
degradation and the expert network by analyzing the IC curves of the dataset.

*“ **Supplementary Figure 28. IC Aging Curve Analysis of the Dataset***

*This section aims to demonstrate the rationality and universality of dividing battery aging into*
*multiple stages using IC curves, and to explain how these parameters reveal the battery degradation*
*mechanisms and reflect their physical significance. This assumption is based on numerous studies*
*on lithium battery degradation patterns. For instance, existing research suggests that during the*
*battery's lifecycle, the formation and thickening of the SEI layer affect the capacity degradation in*
*the early stages, while lithium-ion deposition typically occurs in the later stages of battery use and*
*is one of the main causes of the sharp capacity decline.*

*IC curves reveal internal changes in the battery by showing the subtle variations between battery*
*capacity and voltage, especially during the aging process, where the SEI layer growth, lithium-ion*
*loss, and internal electrochemical reactions significantly impact the IC curve. As shown in the figure,*
*different colors represent different SOH samples. In the early charging and discharging cycles, the*
*curve displays sharp peaks, indicating the battery is in a healthy state with significant and rapid*
*capacity changes, mainly reflecting the formation of the SEI layer and surface chemical reactions.*

*As the cycle count increases, the peaks of the IC curve become more rounded and shift to the right,*
 *and longer flat sections appear, indicating that the battery's reversible capacity is gradually*
 *decreasing. When the battery enters a more severe degradation phase, the peaks of the IC curve*
 *gradually decrease. At this point, degradation mechanisms such as lithium-ion deposition and loss*
 *of active materials dominate the capacity decline. The widening of the peaks in the curve and the*
 *sharp capacity drop indicate that the battery's energy storage capacity is severely compromised.*

 By directly performing t-SNE clustering on the weights of different experts, PIMOE shows clear
 clustering boundaries for samples from different degradation stages, further demonstrating the
 effectiveness of the MoE architecture in identifying different degradation stages.

*“Supplementary Figure 19. t-SNE Dimensionality Reduction Visualization of Expert Weights from*
 *NCA Battery Test Model.*

*For the NCA-material test batteries under different operating conditions in the UL dataset, we*
 *randomly selected a pre-trained model and used full-lifecycle single-cycle test samples as input.*
 *After t-SNE clustering, the expert weights output by the model were visualized, with samples from*
 *different aging stages color-mapped accordingly (see Supplementary Note 8 for details). In*
 *Supplementary Figures 1-6, as well as through SHAP and Spearman correlation coefficient analysis,*
 *the features input into the AMDP module show strong correlation with the degradation stages*

themselves. In the two-dimensional latent space after clustering, there is a clear boundary between
healthy and aged batteries, further demonstrating the effectiveness of the MOE architecture in
identifying different degradation stages and validating the rationale for using expert weights in
second-life utilization decisions. Figures (a)-(d) represent three different operating conditions. “

The authors have made the following modifications to the discussion:

“PIMOE model shows great potential to transform the battery reusing and recycling industry
landscape by reducing the need of manual-assisted testing approach to an automated data-driven
decision support system. However, it must be acknowledged that PIMOE approach primarily relies
on electrochemical characteristics (such as voltage-capacity relationships and relaxation voltage)
for macroscopic diagnostics. While extensive validation confirms the physical understanding of
these features^{13,33}, they essentially remain statistical correlations instead of physical causality. It is
recommended that future work should focus on non-invasive in-situ sensing signals^{26,27,34}, such as
vibration sensing, strain sensing, ultrasonic signals, fiber optic sensing to enhance internal state
observation for a physics-informed modeling, instead of physically interpretable presented in this
work. Building a "physics-statistics" fusion framework has the potential to improve the
interpretability of PIMOE²⁹, for example, parameters from battery electrochemical or equivalent

*circuit models could be updated online as an intermediate physical interpretation layer for*
*degradation trajectory prediction. Integrating primary governing equations of battery modeling*
*with data-driven methods would also provide physical interpretability³⁵.*

”

**Comment 5**

Dataset. The authors should make it clearer that the aging datasets used in this work are obtained
from the literature and provide the corresponding references.

**Response to Comment 5**

Thank you for the comment. The authors have clarified in both the main text and supplementary
materials that the aging datasets used in this study are sourced from the referenced literature, and
the corresponding references have been provided.

**Comment 6**

Typos and grammatical mistakes

• Line 56: relavent → relevant

• Line 65: Larbor → labor

• Line 65: awared → aware

• Line 79: the demanding → demanding

• Line 95: implication → implication

• Line 123: "uncertain secondary use condition" → "uncertain secondary use conditions"

• Line 279-280: "a interpretability" → "an interpretability"

• Line 389: exhitbit → exhibit

• Line 399: extenstive → extensive.

**Response to Comment 6**

Thank you for the comment. The authors have carefully revised our manuscript to address the issues
you raised. The revised manuscript has been significantly elaborated, revised, and polished.

**Response to Reviewer #4**

Authors describe a method for estimation of Li-ion cell capacity and combine it with predictive
model that allows forecasting of future capacity degradation trends. Authors claim that the proposed
method allows history-free estimation of State-of-Health (SOH) of the battery hence eliminating a
need for historical usage records or extensive lab tests. Elimination of data requirements is a very
valuable technical goal since unavailability of historical usage or test data is a biggest obstacle for
accurate estimation of SOH.

Dear Reviewer,

The authors respond to your suggestions point by point and carefully revise the manuscript and
supplementary information, where you can find corresponding revisions using “*track changes*”
mode. Changes are colored in *blue* in revised clean version. The authors truly hope that the responses
appropriately address your justified concerns, and that revised manuscript meets your expectations.

**Comment 1**

While authors claim history-free method, the training process used for training of the estimation and
prediction components of the proposed model relies on extensive test data collected offline. I think
that authors need to revise some of their claims related to this discrepancy. That training data is the
missing ‘history’. It is very well known that with sufficiently rich degradation data collected offline
one can train a data-driven model to achieve pretty much any desired accuracy.

**Response to Comment 1**

Thank you for the comment. The PIMOIE indeed relies on offline testing data from laboratory
experiments for initial training. This characteristic stems from the fundamental need for high-quality
labeled data in deep learning methods. The training data comes from battery full lifecycle testing
conducted in laboratory environments, and the statement about not requiring historical data can
indeed cause confusion for readers. The authors have revised the paper in the following aspects:

- ● The authors have revised the abstract to emphasize that historical data is not required in
deployment scenarios, thereby avoiding potential confusion for readers.
- ● The authors made modifications in the Broader Context to emphasize that our approach has
lower data requirements and higher applicability in practical applications.
- ● The authors have revised the introduction to explain the challenges of predicting degradation
trajectories in scenarios where historical data are unavailable.

● The authors clarified in the main text the different data requirements for the two phases: offline
training and online application.

● The authors have added a deeply degraded battery dataset to the experimental design to
demonstrate that the proposed method does not rely on historical usage conditions.

We have made modifications in the Broader Context.

*“As electric vehicle batteries reach end-of-life, their residual capacity presents substantial*
*opportunities for second-life applications. However, ensuring safety and sustainable reusing*
*remains challenging due to the absence of historical data and the uncertainty of future operating*
*conditions. Conventional models often rely on consistent use profiles and complete cycling histories,*
*which are rarely available in real-world practice. ”*

We have made modifications to the abstract.

*“Retired electric vehicle batteries offer immense potential to support energy infrastructure stability*
*in underdeveloped regions as second-life use, but uncertainties in battery degradation behaviors*
*pose major safety concerns. This work proposes a Physics-interpretable Mixture of Experts (PIMOE)*
*network that predicts battery degradation trajectories using partial, field-accessible signals in a*
*single cycling operation. ”*

[revised manuscript text omitted]

We have added the following description in the Model performance and generalization capability:

“Fig. 3b reveals that in TPSL data, baseline methods displayed limited capability in capturing
 long-term trends after future load changes and completely failed during initial load transitions. In
 contrast, PIMOIE effectively identified transitional characteristics between phases through future
 usage condition integration. Supplementary Figs.11-14 further demonstrate PIMOIE's successful
 learning of relationships between future operating conditions and capacity performance. In the LSD
 dataset, the baseline models exhibited larger errors in predicting the later stages of degradation
 trajectories. This is mainly due to the unknown and varying operating conditions in the first phase,

which resulted in highly heterogeneous degradation behaviors during deep aging. In contrast, our
proposed method, benefiting from the AMDP module, achieved more stable and reliable prediction
performance.

[revised manuscript text omitted]

We have added the following description in the Supplementary Table 2:

“**Supplementary Table 2. Technical Specifications of LSD Dataset.**

The table presents the materials, operating conditions, nominal capacities, cutoff voltages, and
 number of batteries used in our experimental dataset. During phase I, as the SOH of the batteries
 declined from 100% to 80%, 86 cells were divided into 16 groups, with each group subjected to a
 distinct charge–discharge protocol. In phase II, as the SOH further decreased from 80% to 50%, all
 86 cells were cycled under a unified protocol. The dataset is designated as the LSD Dataset.

In the phase 2 dataset, although all batteries were operated under identical cycling conditions,
 differences in the loading conditions during their first-life usage led to varying impacts on internal
 degradation mechanisms, resulting in significantly divergent degradation trajectories during the
 second-life phase. This phenomenon highlights the profound influence of historical usage conditions
 on the long-term health evolution of batteries and provides an ideal validation scenario for
 evaluating the model’s predictive capability under unknown historical conditions and deep
 degradation states. We utilized only the second-phase dataset to simulate the model’s ability to
 predict degradation trajectories of deeply aged, retired batteries in deployment scenarios where
 historical data are unavailable.

Cell types	Material	Working conditions			Nominal capacity (Ah)	Cutoff voltage (V)	Number of cells
		Charge rate	Discharge rate	Temperature (°C)			
Phase 1	LiCoO ₂ and LiNi _{0.5} Co _{0.2} Mn _{0.3} O ₂ / graphite anodes	0.5	1	25	2.4	3.0-4.2	8
		0.5	1	25			8
		0.5	1	25			6
		1	1	25			8
		1.5	1	25			8
		2	1	25			4
		0.5	2	25			4
		0.5	3	25			4
		1	1	25			4
		1	1	25			4
1.5	1	25	4				

	1.5	1	25	4
	2	3	25	4
	0.5	0.2	25	3
	0.5	0.5	25	7
	0.5	2	25	4
Phase 2	0.5	1	25	86

**Comment 2**

I recommend the authors to put a bit more effort in connecting the proposed modeling method to
 physics of Li cell degradation. Without this connection between selected ML features and vaguely
 mentioned degradation modes the paper does not stand out from a long list of data-driven modeling
 papers for SOH estimation.

**Response to Comment 2**

Thank you for the comment. Although different degradation stages exhibit distinct characteristics,
 their corresponding electrochemical behaviors follow relatively unique yet universally observable
 patterns. These features can be extracted through physics-informed feature engineering, while the
 MOE network decomposes the complex task into sub-tasks handled by specialized expert modules
 capable of capturing multi-level and nonlinear feature representations. The concept of decomposing
 a complex problem into multiple sub-tasks inherently aligns with the coupled nature of degradation
 mechanisms in batteries. The authors have revised the paper in the following aspects:

- ● The authors have revised the title of the manuscript to avoid potential ambiguity.
- ● The authors have revised the introduction to further discuss physical modeling methods
- ● The authors further illustrate the correlation between degradation mechanisms at different
 stages of degradation and the expert network by analyzing the IC curves of the dataset.
- ● The authors further validated the rationality of these physical features through SHAP analysis
 and Spearman correlation coefficients.
- ● The authors have revised the discussion section to clearly outline the current limitations of the
 method and the direction for future improvements:

The authors have revised the title of the paper:

*“PIMOE: Physically interpretable mixture of experts network for battery degradation trajectory*
 *prediction amid second-life complexities.”*

The authors have revised the introduction to further discuss physical modeling methods:

[revised manuscript text omitted]

The authors further illustrate the correlation between degradation mechanisms at different stages of
degradation and the expert network by analyzing the IC curves of the dataset.

***" Supplementary Figure 28. IC Aging Curve Analysis of the Dataset***

*This section aims to demonstrate the rationality and universality of dividing battery aging into*
*multiple stages using IC curves, and to explain how these parameters reveal the battery degradation*
*mechanisms and reflect their physical significance. This assumption is based on numerous studies*

on lithium battery degradation patterns. For instance, existing research suggests that during the
 battery's lifecycle, the formation and thickening of the SEI layer affect the capacity degradation in
 the early stages, while lithium-ion deposition typically occurs in the later stages of battery use and
 is one of the main causes of the sharp capacity decline.

IC curves reveal internal changes in the battery by showing the subtle variations between battery
 capacity and voltage, especially during the aging process, where the SEI layer growth, lithium-ion
 loss, and internal electrochemical reactions significantly impact the IC curve. As shown in the figure,
 different colors represent different SOH samples. In the early charging and discharging cycles, the
 curve displays sharp peaks, indicating the battery is in a healthy state with significant and rapid
 capacity changes, mainly reflecting the formation of the SEI layer and surface chemical reactions.
 As the cycle count increases, the peaks of the IC curve become more rounded and shift to the right,
 and longer flat sections appear, indicating that the battery's reversible capacity is gradually
 decreasing. When the battery enters a more severe degradation phase, the peaks of the IC curve
 gradually decrease. At this point, degradation mechanisms such as lithium-ion deposition and loss
 of active materials dominate the capacity decline. The widening of the peaks in the curve and the
 sharp capacity drop indicate that the battery's energy storage capacity is severely compromised.”

In Supplementary Figures 1-6, we directly display the correlation between these physical features
and battery aging. Furthermore, we have further validated the rationality of these physical features
through SHAP analysis and Spearman correlation coefficients.

The authors have made the following modifications to the Supplementary Figure 22:

*“ To quantify the influence of input features on the model's predictions and elucidate the*
*underlying physical mechanisms, we employed SHapley Additive exPlanations (SHAP) analysis and*
*Spearman's rank correlation analysis. As shown in Fig. a, the SHAP analysis based on NCA+NCM*
*battery data reveals the differential contributions of the 12 physics-informed features to the final*
*prediction outcome. Concurrently, Spearman's correlation analysis in Fig. b demonstrates that most*
*features exhibit statistically significant correlations with SOH. These two analytical results*
*corroborate each other, collectively confirming the robustness and reliability of the selected physical*
*features in effectively characterizing the battery degradation state.*

Based on the strong correlation between the selected input features of the AMDP module and SOH,
the degradation gating network can effectively distinguish data samples from different degradation
stages to make corresponding predictions. By directly performing t-SNE clustering on the weights
of different experts, PIMOE shows clear clustering boundaries for samples from different
degradation stages, further demonstrating the effectiveness of the MoE architecture in identifying
different degradation stages.

*“Supplementary Figure 19. t-SNE Dimensionality Reduction Visualization of Expert Weights from*
*NCA Battery Test Model.*

*For the NCA-material test batteries under different operating conditions in the UL dataset, we*

randomly selected a pre-trained model and used full-lifecycle single-cycle test samples as input.
After t-SNE clustering, the expert weights output by the model were visualized, with samples from
different aging stages color-mapped accordingly (see Supplementary Note 8 for details). In
Supplementary Figures 1-6, as well as through SHAP and Spearman correlation coefficient analysis,
the features input into the AMDP module show strong correlation with the degradation stages
themselves. In the two-dimensional latent space after clustering, there is a clear boundary between
healthy and aged batteries, further demonstrating the effectiveness of the MOE architecture in
identifying different degradation stages and validating the rationale for using expert weights in
second-life utilization decisions. Figures (a)-(d) represent three different operating conditions. “

The authors have revised the discussion section to clearly outline the current limitations of the
method and the direction for future improvements:
“PIMOE model shows great potential to transform the battery reusing and recycling industry
landscape by reducing the need of manual-assisted testing approach to an automated data-driven
decision support system. However, it must be acknowledged that PIMOE approach primarily relies
on electrochemical characteristics (such as voltage-capacity relationships and relaxation voltage)
for macroscopic diagnostics. While extensive validation confirms the physical understanding of

*these features^{13,33}, they essentially remain statistical correlations instead of physical casualty. It is*
*recommended that future work should focus on non-invasive in-situ sensing signals^{26,27,34}, such as*
*vibration sensing, strain sensing, ultrasonic signals, fiber optic sensing to enhance internal state*
*observation for a physics-informed modeling, instead of physically interpretable presented in this*
*work. Building a "physics-statistics" fusion framework has the potential to improve the*
*interpretability of PIMOE²⁹, for example, parameters from battery electrochemical or equivalent*
*circuit models could be updated online as an intermediate physical interpretation layer for*
*degradation trajectory prediction. Integrating primary governing equations of battery modeling*
*with data-driven methods would also provide physical interpretability³⁵. ”*

**Comment 3**

Line 130: Please, explain what the Trend vector is exactly. Is it a vector of capacity or SOH values
over one cycle vs time? Several cycles?

**Response to Comment 3**

Thank you for the comment. The trend vector is a latent feature representation generated from field-
acquired current cycle data, designed for predicting capacity degradation over multiple future cycles.
In most conventional methods, the output of this vector is directly used as the prediction value.
However, the authors argue that incorporating future load conditions is essential for meaningful
real-world lifespan prediction, as future operating conditions significantly influence capacity
degradation. In the original design, this vector does not account for the impact of future load
conditions on degradation trajectories; rather, it reflects a preliminary prediction of future
degradation based solely on the current health state. The length L of the trend vector strictly
corresponds to the number of prediction cycles.

The authors have provided further explanation and visualization in the supplementary Figure 25.:

“ **Supplementary Figure 25. Visualizations of the trend vectors across different cycles in various**
*datasets*

*This section aims to provide a clearer explanation of the Trend vector we have defined, with*
*visualizations of the trend vectors across different cycles in various datasets. The trend vector is a*
*latent feature representation generated from field-acquired current cycle data, designed for*
*predicting capacity degradation over multiple future cycles. In most conventional methods, the*
*output of this vector is directly used as the prediction value. However, we argue that incorporating*

*future load conditions is essential for meaningful real-world lifespan prediction, as future operating*
 *conditions significantly influence capacity degradation. In the original design, this vector reflects a*
 *preliminary prediction of future degradation based solely on the current health state, without*
 *accounting for the impact of future load conditions on degradation trajectories. The length L of the*
 *trend vector strictly corresponds to the number of prediction cycles. In the FORNN module, we*
 *integrate the trend vector with the planned future load on a cycle-by-cycle basis according to the*
 *prediction horizon, enabling the model to simultaneously achieve accurate assessment of the current*
 *state and embedding of the influence of future operating conditions. Different colors represent*
 *different cycle numbers. “*

**Comment 4**

It would be very valuable to a reader to see an explanation of why/how selected AMDP features
 contain information about SOH of the cell. Without this info a reader stays uneducated about why
 the proposed method should work and paper reads just like yet another ‘we stacked several MLMs
 to achieve good KPIs’ paper.

**Response to Comment 4**

Thank you for the comment. First, the physical features we selected have been widely demonstrated
 in previous studies to be strongly correlated with degradation mechanisms, as shown in
 Supplementary Figures 16–18. Second, as mentioned in your Comment 3, the magnitude of the
 trend vector exhibits a clear correlation with the number of cycles across different samples. Finally,
 the authors provide direct evidence showing that for samples with the same SOH but from different

batteries, the model automatically selects specific types of expert networks for prediction. The
authors have revised the paper in the following aspects:

- ● The authors have updated Supplementary Figures 16–18 in the revised manuscript to provide
a more comprehensive presentation and analysis of this phenomenon.
- ● The authors have revised the supplementary Figure 15 to provide a more comprehensive
analysis of this phenomenon.

The authors have updated Supplementary Figures 16–18 in the revised manuscript to provide a more
comprehensive presentation and analysis of this phenomenon.

*Supplementary Figure 18. Expert weights output by the model for an entire batch of NCA-material*
*batteries retired at different SOH levels.*

*Visualization of expert weights output by the model for an entire batch of batteries retired at different*
*SOH levels during practical deployment. For all NCA-material batteries in selected SOH retirement*
*scenarios, we randomly chose one pre-trained model to visualize the expert weights across the entire*
*battery population. The horizontal axis represents the number of retired batteries, while the vertical*
*axis indicates the weights assigned by five corresponding experts, with color intensity reflecting*
*weight values Fig. (a)-(e) represent three distinct operating conditions: 25-1-1,25-05-1,25-025-*
*1,35-05-1,45-05-1 respectively.*

The authors have revised supplementary **Figure 15** to provide a more comprehensive analysis of
 this phenomenon.

“ *Supplementary Figure 15. Correlation between expert weights and degradation stages under
 different operating conditions.*

*Correlation between expert weights and degradation stages under different operating conditions.*

*By randomly selecting pre-trained models corresponding to specific conditions and using single-*

*cycle test samples from the full lifecycle of batteries across different datasets as input, we visualize*

*the relationship between the evolving expert weights in the model output and the degradation stages.*

*The horizontal axis represents test samples from different degradation stages, while the vertical axis*

indicates the weights assigned by five experts to each sample, with color intensity reflecting the
 magnitude of expert weights. As degradation progresses, different experts dominate corresponding
 stages, enabling adaptive integration of expert weights to predict degradation trajectories for
 samples at various degradation phases. “

Furthermore, as mentioned in your Comment 2, The authors further demonstrated the strong
 correlation between the AMDP features and battery aging through IC curve analysis, SHAP and
 Spearman correlation analyses, as well as t-SNE clustering of expert weights.

**Comment 5**

Figure 2c and subsequent figures show fluctuations in measured capacity by double-digit percent.
 How is this possible? How was exactly capacity measured in experiments?

**Response to Comment 5**

Thank you for the comment. During the second life phase of the TPSL dataset, CC charging is
employed. Variations in charging current magnitudes lead to differences in the maximum available
capacity per cycle, as well as divergent degradation rates. This serves as the primary cause for the
double-digit percentage fluctuations observed in capacity measurements. In this study, the true
capacity labels correspond to the maximum discharge capacity of the current cycle.

**Comment 6**

There is a large drop in capacity from First to Second life on Figure 2c. Why is there such a drop?

**Response to Comment 6**

Thank you for the comment. The original dataset employed a fixed charge-discharge protocol with
CCCV charging during the first-life stage, while switching to CC charging in the second-life stage.
This change in charging protocols resulted in significant capacity degradation after different
charging conditions were applied. The authors acknowledge that the original description may have
caused confusion among readers.

The authors have made the following modifications to the Supplementary Table 3:

*“ The table presents the materials, operating conditions, nominal capacities, cutoff voltages, and
number of batteries used in our experimental dataset. After the initial 20 cycles, the load conditions
were modified for these batteries, with 22 batteries undergoing different constant operating
condition cycles and 55 batteries subjected to random operating condition cycles. The dataset is
designated as the TPSL Dataset.*

*In this study, "TPSL-Random" refers to batteries operating under random conditions during their
second-life usage, while "TPSL-Fixed" indicates batteries operating under fixed conditions during
their second-life usage. The TPSL Dataset encompasses a total of 66 second-life operating
conditions (55 random and 11 fixed) across 77 batteries. In the TPSL dataset, the charging protocol
was changed from the CCCV method used during the first life to the CC method employed in the
second-life stage. This shift in operating conditions resulted in a significant sudden capacity drop
between the cycles adjacent to the transition point. In the TPSL-Fixed dataset, batteries under
different fixed conditions exhibited significantly divergent degradation trajectories. In the TPSL-
Random dataset, the random operating conditions resulted in notable maximum capacity differences
even between adjacent cycles. This further underscores the necessity of incorporating future*

*operating conditions as conditional input. Notably, for each sample, we utilize only the current*
 *cycle's data without requiring any historical cycle information.*

Cell type	Material	Working conditions			Nominal capacity (Ah)	Cut-off voltage (V)	Number of cells
		Charge C rate	Discharge C rate	Temperature (° C)			
		2	1	25			3
		3	1	25			3
		1	2	25			4
		2	2	25			3
		3	2	25			3
NC	LiNi0.5Co0.2Mn0.3	2	3	25	2.4	3.0-	3
M	O2 / graphite anodes	3	3	25			4.2
		Random current (1C~3C)	3	25			55

**Comment 7**

Figure 3b, and 2c show not only drops but also jumps up to 20% in measured capacity. Can you,
 please provide explanation of what is going on there? This is not physical behavior for NCM/NCA
 cells.

**Response to Comment 7**

Thank you for the comment. The capacity surge you mentioned is indeed present in the dataset. As
 noted in your Comment #5, the TPSL dataset incorporates 56 types of randomized charge-discharge
 protocols to simulate diverse secondary-use conditions characterized by high complexity and
 stochasticity. These varying protocols lead to significant differences in the maximum available
 capacity even for the same battery across consecutive cycles. This variation occurs because the CC
 charging method, when applied at different current rates, results in different maximum charge levels
 achievable. This inherent variability invalidates methods that assume consistent future operating
 conditions, such as classical time-series models. In real-world secondary applications of retired
 batteries, however, future load profiles are often unpredictable, further highlighting the complexity
 of task.

**Comment 8**

Line 199-200. Capacity knee at EOL of a cell, ‘third phase’, is not always related to Li-plating.

**Response to Comment 8**

Thank you for the comment. The original text contained overly simplified and absolute expressions,
which did not fully reflect the complexity of battery degradation mechanisms. Not all capacity
inflection points are directly related to lithium-ion deposition. In some cases, the inflection points
of capacity decline may also be associated with other degradation mechanisms, such as SEI film
thickening or the loss of active materials, which influence the battery's capacity retention and
degradation rate at different stages. Therefore, in the revised version, The authors further clarify this
point and emphasize that the battery degradation process is multi-faceted, and degradation modes
at each stage may interact, leading to different capacity changes.

*“In Fig. 3a, existing research demonstrates that battery degradation can be divided into three*
*typical while distinct degradation phases: SEI formation, SEI thickening, and lithium plating*^{1,45}.
*During the battery's lifecycle, the formation and thickening of the SEI layer impacts the battery's*
*capacity degradation in early stages, while lithium-ion deposition typically occurs in the later stages*
*of the battery's use and is one of the main causes of sharp capacity decline. However, it is important*
*to note that the capacity knee at the end of life (EOL) is not always directly related to lithium plating.*
*In some cases, the capacity decline at the EOL may also be influenced by other degradation*
*mechanisms, such as SEI thickening, loss of active material, or electrolyte degradation, which can*
*also contribute to the sharp capacity drop observed in later stages. The degradation patterns across*
*these phases exhibit significant differences in both available capacity and subsequent aging rates,*
*especially in the slope of the degradation trajectory, with each phase reflecting distinct degradation*
*behaviors.*^{46,47}. *Supplementary Figure 28 illustrates that the definition of degradation stages and*
*mechanisms can be regarded as typical and reasonable. ”*

**Comment 9**

Line 207. Again, not clear why measured capacity jumps around and why the model follows this
trend.

**Response to Comment 9**

Thank you for the comment. As noted in your Comments #5 and #6, the TPSL dataset exhibits
significant sudden capacity drops as well as frequent capacity surges during the second-life phase.

This phenomenon occurs because the CC charging protocol used in second-life applications, when
applied at varying current rates, leads to substantial variations in the maximum available capacity
1913 per cycle—even between adjacent cycles. The PIMOE model proposed in this study achieves robust
prediction performance despite these condition-induced capacity fluctuations, which is primarily
attributed to its carefully designed FORNN module. The key innovation of FORNN lies in its
explicit concatenation of assumed operational conditions for each future cycle with the trend vector,
which together serve as the input to the recurrent network. This design enables the model to
dynamically simulate the impact of diverse usage conditions on the degradation trajectory, thereby
achieving "condition-dependent" prediction.

Comment 10

Line 243. I think the conclusion about use of higher SOC range is a bit ML model development
centric. A physical reason could be that the snippet of data starting at SOC > 50% simply does not
contain enough information about OCV features that can help the model to distinguish between
components of degradation such as Li loss vs AML.

**Response to Comment 10**

Thank you for the comment. The data in the high SOC range may lack sufficient open-circuit voltage
feature information, which could impair the model's ability to distinguish between degradation mode
components. This perspective enhances analysis from the standpoint of physical mechanisms. The
authors have revised the paper in the following aspects:

- ● The authors have revised the main text to explain that the poorer prediction results in the high
SOC range may be due to the lack of sufficient open-circuit voltage feature information.
- ● The authors have further discussed future work in the discussion section, focusing on
predicting degradation trajectories of retired batteries with random initial SOC.

The authors have added the following in the main text:

*“ The observed performance disparity can be attributed not only to the reduced volume of training
data available at higher initial SOC from a machine learning perspective but also, more
fundamentally, to the inherent lack of adequate OCV feature information in high-SOC data
segments from a physicochemical standpoint, which hinders the model's ability to differentiate
degradation modes such as lithium inventory loss and active material loss (LAM). Furthermore, as
the computations for the UL dataset do not involve cycle-level condition changes, the model*

*outcomes rely more heavily on the data volume at the initial SOC, resulting in more pronounced*
*performance degradation as the initial SOC increases. “*

The authors have made modifications to the discussion.

*“ Another aspect is that random initial SOC is an important factor in the data availability of*
*retired batteries. Although the effectiveness of PIMOIE in predicting degradation trajectory under*
*different initial SOC conditions has been validated, the initial SOC of retired batteries is more*
*uncontrollable due to the time-consuming nature of acquisition such information. Future work*
*suggests considering SOC normalization during the feature extraction phase or introducing an*
*SOC-aware routing mechanism in the model, to better handle battery degradation difference under*
*different SOC states. Although the paper has considered four very different application conditions,*
*namely UL, LSD, TPSL-Random, TPSL-fixed, given that the second-life application of the battery*
*is safety-critical, more extreme conditions need to be investigated. This is because the robustness of*
*the dataset to non-steady-state use conditions, such as the battery with thermal runaway or internal*
*short circuits, has not been included. Considering the bias of the training data is crucial, and it is*
*necessary to assess the uncertainty and confidence intervals of the predicted degradation*
*trajectories, as UL and TPSL data hardly cover all environmental fluctuations under actual service*
*conditions, such as thermal non-uniformity at the battery pack level. Future work is suggested to*
*collect real-world variables that include different manufacturers, degradation mechanisms, and*
*cell-to-cell variations. ”*

**Comment 11**

Figure 3: What is the prediction horizon? Are we predicting x-number of cycles ahead or the entire
second life until certain SOH?

**Response to Comment 11**

Thank you for the comment. The description of the prediction horizon in the original text may have
caused confusion among readers. The PIMOIE model proposed in this study is designed to achieve
long-term prediction targeting a fixed number of future cycles. In standard experiments, to balance
prediction accuracy and engineering practicality, we adopted 50 cycles as the benchmark prediction
length (accounting for approximately 30% of the median lifespan of typical batteries). Furthermore,
the method proposed in this study maintains robust performance even when predicting up to 150
cycles, further demonstrating its effectiveness in extending the prediction horizon to long-term

degradation trajectory forecasting. The authors have revised the content of Supplementary Note 3
to provide a more comprehensive explanation of the format of the results predicted by the model.
*“Supplementary Note 3. Selection of prediction horizons.*
*In practical applications, while extending the prediction horizon of degradation trajectories*
*enhances model utility, excessive prediction spans may compromise accuracy as illustrated in Fig.*
*5d. Nevertheless, our proposed solution maintains robust performance even when predicting 150*
*cycles. To balance prediction accuracy with engineering practicality, this study adopts a 50-cycle*
*prediction horizon as the benchmark (representing approximately 30% of a typical battery's median*
*lifespan). Notably, for special operating conditions like NCA-25-1-1 with shorter cycle lives (~30*
*cycles), the prediction length is correspondingly reduced to 10 cycles to ensure reliability.*
*It should be emphasized that while existing literature reports longer prediction horizons*¹⁴*, such*
*methods typically require fixed sampling intervals and complete historical cycle data as input, a*
*design that limits their deployment in dynamic operating conditions*⁴⁸*. In contrast, our innovative*
*approach requires only partial current cycle data as input and adapts to random initial SOC*
*sampling strategies. This breakthrough not only significantly enhances real-world applicability but*
*also provides technical assurance for end-user deployment convenience.*
*This design philosophy aligns perfectly with industry demands for "plug-and-play" prediction*
*models, achieving an optimal compromise between prediction accuracy and data acquisition*
*simplicity.”*

**Comment 12**

Line 375, Discussion: PIMOE needs reach experimental offline test data to train on according to the
described methodology. Please state that explicitly.

**Response to Comment 12**

Thank you for the comment. Like your comment 1, this method indeed requires offline training data
to establish the initial model. However, it should be noted that during the real-time deployment and
application phase, there is no need to use the historical data of the battery under test. The statement
about not requiring historical data can indeed cause confusion for readers, so The authors have made
the following revision in the main text:

“ Existing research paradigm of degradation trajectory prediction predominantly relies on
historical data access and assumed identical future use conditions, which is often not true for
second-life batteries⁴⁹. This paradigm inflicts either expensive lifecycle data acquisition or time-
consuming capacity calibration tests post-retirement, also raising safety concerns due to its inability
to account for second-life use uncertainty. The proposed PIMOE framework, after offline training,
utilizes data already available in battery management systems in deployment scenarios to directly
predict degradation trajectories under assumed but uncertain future second-life use conditions,
without relying on historical data, eliminating the need for extra offline testing. The success of
PIMOE stems from adaptively and interpretably classifying degradation modes, statistically the
geometric slope of the degradation trajectory at the observation point, to inform use-condition-
dependent degradation trajectory prediction in the extended second-life use. ”

**Comment 13**

Line 461, equation 5: What is exactly the ‘basis response’ in equation 5? What are exactly the
degradation modes?

**Response to Comment 13**

Thank you for the comment. In Equation 5, the "base response" refers to the unweighted degradation
trajectory prediction independently output by each expert network for a given input sample. Its
physical essence is as follows: each expert network is designed to specifically simulate a particular
dominant degradation mechanism. The degradation of each battery is dominated by certain aging
mechanisms, with multiple mechanisms acting collectively. Therefore, the prediction result is a
weighted aggregation of the outputs from different experts, as illustrated in Figure 4a. For different
battery samples, the degradation-router dynamically assigns weights to each expert network. When
the battery SOH is high (e.g., >95%), the model tends to assign higher weights to Expert 1 and
Expert 2, indicating that the prediction mechanisms they dominate are highly correlated with the
primary processes in the early stages of degradation (such as SEI formation and growth). When the
battery SOH is low (e.g., <85%), the weights of Expert 4 and Expert 5 increase significantly,
suggesting that they are more focused on simulating key mechanisms that emerge in the later stages
of degradation (such as lithium metal plating and the accelerated degradation it induces). Thus, the
"base response" essentially represents a "local prediction" from the perspective of a specific
degradation mechanism. Through Equation 6, these local predictions, based on different physical

mechanisms, are synthesized into a "global prediction" that reflects the overall degradation behavior
of the battery under the coupled effects of complex multiple mechanisms.

**Comment 14**

I think if degradation modes are described that may help to explain ‘physics-informed’ part of the
title. Without this explanation I do not see any ‘physical’ components in the proposed model.

**Response to Comment 14**

Thank you for the comment.

The authors have revised the paper in the following aspects:

- ● The authors have revised the title of the manuscript to avoid potential ambiguity.
- ● The authors further illustrate the correlation between degradation mechanisms at different
stages of degradation and the expert network by analyzing the IC curves of the dataset.
- ● The authors further validated the rationality of these physical features through SHAP analysis
and Spearman correlation coefficients.
- ● The authors have revised the introduction to further discuss physical modeling methods
- ● The authors have revised the discussion section to clearly outline the current limitations of the
method and the direction for future improvements:

The authors further illustrate the correlation between degradation mechanisms at different stages of
degradation and the expert network by analyzing the IC curves of the dataset.

*“ Supplementary Figure 28. IC Aging Curve Analysis of the Dataset*

*This section aims to demonstrate the rationality and universality of dividing battery aging into*
*multiple stages using IC curves, and to explain how these parameters reveal the battery degradation*
*mechanisms and reflect their physical significance. This assumption is based on numerous studies*
*on lithium battery degradation patterns. For instance, existing research suggests that during the*
*battery's lifecycle, the formation and thickening of the SEI layer affect the capacity degradation in*
*the early stages, while lithium-ion deposition typically occurs in the later stages of battery use and*
*is one of the main causes of the sharp capacity decline.*

*IC curves reveal internal changes in the battery by showing the subtle variations between battery*
*capacity and voltage, especially during the aging process, where the SEI layer growth, lithium-ion*
*loss, and internal electrochemical reactions significantly impact the IC curve. As shown in the figure,*
*different colors represent different SOH samples. In the early charging and discharging cycles, the*

*curve displays sharp peaks, indicating the battery is in a healthy state with significant and rapid*
 *capacity changes, mainly reflecting the formation of the SEI layer and surface chemical reactions.*
 *As the cycle count increases, the peaks of the IC curve become more rounded and shift to the right,*
 *and longer flat sections appear, indicating that the battery's reversible capacity is gradually*
 *decreasing. When the battery enters a more severe degradation phase, the peaks of the IC curve*
 *gradually decrease. At this point, degradation mechanisms such as lithium-ion deposition and loss*
 *of active materials dominate the capacity decline. The widening of the peaks in the curve and the*
 *sharp capacity drop indicate that the battery's energy storage capacity is severely compromised.*

The authors have made the following modifications to the Supplementary Figure 22:

*“ To quantify the influence of input features on the model's predictions and elucidate the*
 *underlying physical mechanisms, we employed SHapley Additive exPlanations (SHAP) analysis and*
 *Spearman's rank correlation analysis. As shown in Fig. a, the SHAP analysis based on NCA+NCM*
 *battery data reveals the differential contributions of the 12 physics-informed features to the final*
 *prediction outcome. Concurrently, Spearman's correlation analysis in Fig. b demonstrates that most*
 *features exhibit statistically significant correlations with SOH. These two analytical results*
 *corroborate each other, collectively confirming the robustness and reliability of the selected physical*
 *features in effectively characterizing the battery degradation state.*

However, these parameters are generated solely through electrical testing and statistical data. To
 avoid any potential confusion for readers, the authors have revised the title of the paper.

*“Physically interpretable mixture of experts (PIMOE) network for battery degradation trajectory*
 *prediction amid second-life complexities.”*

The authors have revised the introduction to further discuss physical modeling methods:

[revised manuscript text omitted]

The authors have revised the discussion section to clearly outline the current limitations of the
method and the direction for future improvements:

*“PIMOE model shows great potential to transform the battery reusing and recycling industry*
*landscape by reducing the need of manual-assisted testing approach to an automated data-driven*
*decision support system. However, it must be acknowledged that PIMOE approach primarily relies*
*on electrochemical characteristics (such as voltage-capacity relationships and relaxation voltage)*
*for macroscopic diagnostics. While extensive validation confirms the physical understanding of*
*these features^{13,33}, they essentially remain statistical correlations instead of physical casualty. It is*
*recommended that future work should focus on non-invasive in-situ sensing signals^{26,27,34}, such as*
*vibration sensing, strain sensing, ultrasonic signals, fiber optic sensing to enhance internal state*
*observation for a physics-informed modeling, instead of physically interpretable presented in this*
*work. Building a "physics-statistics" fusion framework has the potential to improve the*
*interpretability of PIMOE²⁹, for example, parameters from battery electrochemical or equivalent*
*circuit models could be updated online as an intermediate physical interpretation layer for*
*degradation trajectory prediction. Integrating primary governing equations of battery modeling*
*with data-driven methods would also provide physical interpretability³⁵.*

”

**Comment 15**

Equation 6: what are the units of ‘Trend’? Is it in Ah or normalized SOH value? Is it a vector of
values vs time or delta V? L is the number of samples in a cycle or many cycles?

**Response to Comment 15**

Thank you for the comment. The "trend" vector mentioned in this paper is not a direct sequence
representing normalized SOH values or ampere-hours, but a high-dimensional feature vector
embedded through a deep neural network. It encodes the latent feature representation of the battery’s

current health state and short-term degradation trend. The parameter L represents the dimension of
the embedded vector, and the size of L is the same as the length of the predicted future degradation
trajectory. This design both characterizes the battery's current health state and short-term
degradation trend, while also incorporating the designed FORNN module to introduce future
operating conditions cycle by cycle. The authors have revised the description in the main text to
ensure better understanding for the readers.

*“A two-phase collaborative mechanism achieves deep integration of physical insights and data-*
*driven approaches: (1) AMDP, and (2) FORNN. The AMDP module employs “degradation-router”*
*routing to assign expert network weights based on the physical features, adaptively modeling*
*degradation modes under dominant mechanisms for degradation trend representation, which can*
*be formulated as:*

$$Trend_i = \sum_{j=1}^{N_{expert}} G_j(\mathbf{F}_i) \cdot Expert_j(\mathbf{Q}_i) \quad (1)$$

*where, Trend_i denotes computed degradation trend vector for the i-th battery sample, It is*
*important to note that "Trend" is a high-dimensional feature vector embedded through a deep neural*
*network, encoding the latent feature representation of the battery's current health state and short-*
*term degradation trend. Its length is the same as the degradation trajectory we predict.”*

**Comment 16**

Equation 7: Future load parameters, I(t), T(t). Are these time vectors of dimension N?

**Response to Comment 16**

Thank you for the comment. The authors provide a detailed explanation of the dimensional
definition of these future load parameters and their specific role in the model. In the experiments of
this paper, since the data used are from CCCV or CV charging, the future load parameters are not a
time vector of dimension N. Instead, their length is the same as the predicted degradation trajectory,
and for each cycle, they are constant scalars.

Dimensional definition: Assuming the model predicts the future L cycles, the future load parameters
I(t) and T(t) represent the operating conditions for the next L cycles, with dimensions both being L.
The authors have further modified the "Method" section to ensure the reader's understanding.

*“ Second-life applications involve future load conditions that can alter degradation pace as well*

as the maximum available capacity in the current cycle. To include these influences, we employ a
 recurrent neural network that takes as input AMDP output $Trend_i$ and a set of future load
 parameters:

$$C_i = \left[\left(I_{charge}^{(1)}, I_{discharge}^{(1)}, T^{(1)} \right), \dots, \left(I_{charge}^{(L)}, I_{discharge}^{(L)}, T^{(L)} \right) \right] \quad (7)$$

where $C_i \in \mathbb{R}^{L \times 3}$, $I_{charge}^{(\ell)}$, $I_{discharge}^{(\ell)}$, and $T^{(\ell)}$ represent the charge current, discharge current, and
 temperature in future cycle ℓ . For each future load condition, its length is equal to L , the length of
 the degradation trajectory to be predicted. We concatenate these load vectors with the short-term
 trend from AMDP:

$$X_i = \text{Concat}(Trend_i, C_i) \quad (8)$$

The LSTM processes $X_i \in \mathbb{R}^{L \times 4}$ sequentially to predict the capacity trajectory \hat{S}_i over L future
 cycles:

$$\hat{S}_i = \text{LSTM}(X_i) \quad (9)$$

By iterating cycle by cycle, FORNN accommodates a prediction horizon that the user chooses,
 merging the degradation modes signature from AMDP with the load sequence the battery will
 experience under the second-life complexities and uncertainties.”

**Comment 17**

Equation 10: How are true capacity labels generated?

**Response to Comment 17**

Thank you for the comment. In all datasets used in this study, the true capacity labels correspond to
 the maximum discharge capacity of the current cycle, representing the maximum amount of electric
 charge the battery can deliver under the corresponding discharge conditions.

The authors have made the following modifications to the main text:

“ We train the AMDP and FORNN modules jointly by minimizing a loss function that balances
 trajectory accuracy with degradation router diversity. Let S_i be the ground-truth capacity
 trajectory of length L for sample i , the true capacity label represents the maximum discharge
 capacity of that cycle. and let \hat{S}_i be the corresponding model output. The trajectory fidelity
 objective is

$$\mathcal{L}_{traj} = \frac{1}{N} \sum_{i=1}^N \|\hat{S}_i - S_i\|^2 \quad (10)$$

*where the summation is over the training set of size N .*”

**Comment 18**

Noticed typos:

- Line 89, ‘interatively’

- Line 100, ‘extends’

- Line 170, ‘pecices’

- Line 388, ‘valuse’

**Response to Comment 18**

Thank you for the comment. The authors have carefully revised our manuscript to address the issues

you raised. The revised manuscript has been significantly elaborated, revised, and polished.

**References**

[revised manuscript text omitted]

- Preprint at <https://doi.org/10.48550/arXiv.2409.16040> (2024).
- 42. Wang, Y. *et al.* TimeXer: Empowering Transformers for Time Series Forecasting with
Exogenous Variables. Preprint at <https://doi.org/10.48550/arXiv.2402.19072> (2024).
- 43. Pozzi, A., Incremona, A. & Toti, D. Imitation learning-driven approximation of stochastic
control models. *Appl Intell* **55**, 838 (2025).
- 44. Pozzi, A., Incremona, A. & Toti, D. Neural Network-Based Imitation Learning for
Approximating Stochastic Battery Management Systems. *IEEE Access* **13**, 71041–71052
(2025).
- 45. Schuster, S. F. *et al.* Nonlinear aging characteristics of lithium-ion cells under different
operational conditions. *Journal of Energy Storage* **1**, 44–53 (2015).
- 46. Zhao, M., Zhang, Y. & Wang, H. Battery degradation stage detection and life prediction without
accessing historical operating data. *Energy Storage Materials* **69**, 103441 (2024).
- 47. Desai, T., Gallo, A. J. & Ferrari, R. M. G. Multi timescale battery modeling: Integrating physics
insights to data-driven model. *Applied Energy* **393**, 126040 (2025).
- 48. Hsu, C.-W., Xiong, R., Chen, N.-Y., Li, J. & Tsou, N.-T. Deep neural network battery life and
voltage prediction by using data of one cycle only. *Applied Energy* **306**, 118134 (2022).
- 49. Tao, S. *et al.* Collaborative and privacy-preserving retired battery sorting for profitable direct
recycling via federated machine learning. *Nat Commun* **14**, 8032 (2023).

Responses to Reviewer Comments

Title	PIMOE: Physically interpretable mixture of experts network for battery degradation trajectory prediction amid second-life complexities
Revised Title	iMODEL: Prediction of second-life battery degradation trajectory using interpretable mixture of deep expertized learning
Authors	Xinghao Huang 1, Shengyu Tao 1, 2, 3, *, Chen Liang 1, Yining Tang ⁴ , Jiawei Chen 5, Junzhe Shi 2, Yuqi Li 6, Bizhong Xia 1, *, Guangmin Zhou ¹ , *, Xuan Zhang 1, 7 *
Revised Authors	
Journal	Nature Communications
Manuscript ID	NCOMMS-25-53313A

Table of Contents

Response to Reviewers.....	3
Response to Reviewer #1	3
Response to Comment 1	3
Response to Comment 2	3
Response to Comment 3	3
Response to Comment 4	4
Response to Reviewer #2	4
Response to Reviewer #3	4
Response to Reviewer #4	5
Response to Comment 1	5
Response to Comment 2	7
Response to Comment 3	8

**Response to Reviewers**

**Response to Reviewer #1**

Dear Reviewer,

The authors respond to your suggestions point by point and carefully revise the manuscript and
supplementary information, where you can find corresponding revisions using “*track changes*”
mode. Changes are colored in *blue* in revised clean version. The authors truly hope that the responses
appropriately address your justified concerns, and that revised manuscript meets your expectations.

**Comment 1**

In Supplementary Table 1, please check the material composition typos. The NMC composition is
just copied and pasted from NCA.

**Response to Comment 1**

Thank you for the comment. The authors have corrected the typographical errors regarding the NCM
battery material composition and have performed a thorough review of the entire manuscript.

**Comment 2**

Thanks for adding the LSD dataset to the analysis.

**Response to Comment 2**

Thank you for your valuable comments, which have made a significant contribution to this research.
The authors consider your perspectives to be crucial to the proposed methods in this study. We hope
that the findings of this research can draw adequate attention from the academic community to the
emerging field of retired battery echelon utilization.

**Comment 3**

Regarding the linked aging mechanisms, the identified/mentioned mechanisms (i.e. SEI thickness
growth) should be described using words like 'may be' or 'probably' as they could not be validated
within the scope, although they are proven in other works. The batteries are always different from
one to another and trigger different levels of mechanisms under known circumstances.

**Response to Comment 3**

Thank you for the comment. The authors acknowledge the complexity of various degradation
mechanisms across different batteries. Accordingly, the descriptions regarding definitive
degradation mechanisms have been revised in the main text.

**Comment 4**

The code seems clean but did not run it to verify.

**Response to Comment 4**

Thank you for the comment. The authors have revised the README file instructions to make it
easier for readers to run and reproduce the experimental results.

**Response to Reviewer #2**

I appreciate the thorough revision. All my previous concerns have been fully addressed, and I have
no further comments. The code package appears to be well organized and adequately documented.
It includes a clear README. I was able to run the code without particular difficulties, and the results
align with those reported in the manuscript.

Dear Reviewer,

Thank you for your valuable comments, which have made a significant contribution to this
research. The authors consider your perspectives to be crucial to the proposed methods in this study.
We hope that the findings of this research can draw adequate attention from the academic
community to the emerging field of retired battery echelon utilization.

**Response to Reviewer #3**

The authors have delicately revised the paper based on my comments and I am satisfied with the
revised version. As such, I recommend the publication of this work.

Dear Reviewer,

Thank you for your valuable comments, which have made a significant contribution to this
research. The authors consider your perspectives to be crucial to the proposed methods in this study.
We hope that the findings of this research can draw adequate attention from the academic
community to the emerging field of retired battery echelon utilization.

**Response to Reviewer #4**

I would like to thank the authors for putting the effort in addressing reviewer's questions and
suggestions. The manuscript has improved, and most important technical aspects of the proposed
method are now more readable and easier to understand. I think the results are valuable and should
be published.

Dear Reviewer,

Thank you for your valuable comments, which have made a significant contribution to this
research. The authors consider your perspectives to be crucial to the proposed methods in this study.
We hope that the findings of this research can draw adequate attention from the academic
community to the emerging field of retired battery echelon utilization.

The authors respond to your suggestions point by point and carefully revise the manuscript and
supplementary information, where you can find corresponding revisions using "track changes"
mode. Changes are colored in *blue* in revised clean version. The authors truly hope that the responses
appropriately address your justified concerns, and that revised manuscript meets your expectations.

**Comment 1**

My main concern is still about the claim of physical interpretability claimed in the title and abstract.
From my point of view the proposed method does not really provide any physical interpretability of
the results. Weight assignments by PIMOE experts are a result of ML model training. The model
simply learns to differentiate between different capacity degradation trends provided in the training
data. The model uses its available flexibility to put more weight on one part of sub-model closer to
BOL and more weight on another part of sub-model later in cells' life. There is nothing physical
about it, it is simply a result of a properly setup optimization problem. Fig 30 in supplemental
material and its notes does not really tell a reader about physical changes in the cells: it does not
show how much Li or active materials have been lost or how kinetic parameters of the cells have
changed. The way it is presented, it is just a collection of curves that ML model learned to map to
target values.

I suggest keeping the references to physical interpretations in the main text and supplemental
materials as is, but to modify the title and remove the corresponding claim from the abstract and the
introduction. It will be fairer to a reader and avoid any confusion.

**Response to Comment 1**

Thank you for the comment. The authors have made the following revisions:

- ● The authors have revised the title of the manuscript.
- ● The authors have updated the relevant expressions in the Abstract.

● The authors have refined the descriptions and phrasing in the Introduction.

The authors have revised the title of the manuscript.

“ *iMODEL: Prediction of second-life battery degradation trajectory using interpretable mixture*
*of deep expertized learning*”

The authors have updated the relevant expressions in the Abstract.

[revised manuscript text omitted]

**Comment 2**

Supplementary note 5 refers to figure 4e. Could not find anything related to Trend on Fig 4e of the
article.

**Response to Comment 2**

Thank you for the comment. The authors have corrected the figure citation in Supplementary Note
5, changing Fig. 4e to Fig. 4g.

*“We define a vector Trend to represent the output of the AMDP module and visualize the hidden*
*outputs of input samples across the full lifecycle to explain the performance improvement of*
*iMODEL. In Fig.4g, using the NCA-35-05-1 battery, we normalize each value of the Trend vector*
*across different samples for clearer interpretation, thereby demonstrating how the model internally*

*processes samples at different aging stages and clarifying its contribution to degradation trajectory*
*prediction. For the input X of the iMODEL model, the Trend is obtained through the AMDP module.”*

**Comment 3**

Tables 1,2,3 in supplementary material pdf are not properly formatted, difficult to read.

**Response to Comment 3**

Thank you for the comment. The authors have double-checked all PDF documents and implemented
the required changes.